# The quantum-optical nature of high harmonic generation

Alexey Gorlach[1], Ofer Neufeld[1], Nicholas Rivera[2], Oren Cohen[1] & Ido Kaminer[1✉]

High harmonic generation (HHG) is an extremely nonlinear effect generating coherent broadband radiation and pulse durations reaching attosecond timescales. Conventional models of HHG that treat the driving and emitted fields classically are usually very successful but inherently cannot capture the quantum-optical nature of the process. Although prior work considered quantum HHG, it remains unknown in what conditions the spectral and statistical properties of the radiation depart considerably from the known phenomenology of HHG. The discovery of such conditions could lead to novel sources of attosecond light having squeezing and entanglement. Here, we present a fully-quantum theory of extreme nonlinear optics, predicting quantum effects that alter both the spectrum and photon statistics of HHG, thus departing from all previous approaches. We predict the emission of shifted frequency combs and identify spectral features arising from the breakdown of the dipole approximation for the emission. Our results show that each frequency component of HHG can be bunched and squeezed and that each emitted photon is a superposition of all frequencies in the spectrum, i.e., each photon is a comb. Our general approach is applicable to a wide range of nonlinear optical processes, paving the way towards novel quantum phenomena in extreme nonlinear optics.

[1] Technion—Israel Institute of Technology, 3200003 Haifa, Israel. [2] Massachusetts Institute of Technology, Cambridge, MA 02139, USA.
✉email: kaminer@technion.ac.il

High harmonic generation (HHG) is a physical effect that occurs when an atomic, molecular, or solid system is placed in a strong driving laser field and emits photons at frequencies of integer multiples of the driving field frequency[1–3]. HHG provides a coherent source of extreme ultraviolet (EUV) emission and has also paved the way to the field of attoscience[4,5]. This intriguing process has been under investigation for several decades, and it is well-described by a so-called three-step model[2,6,7]. According to this model, the electron (i) tunnels out from the atomic potential suppressed by the intense driving field, (ii) is consequently accelerated in the continuum by the driving field, and (iii) under certain conditions can return to the ion and recombine, emitting a high energy photon. This process repeats itself periodically in time, resulting in a comb emission in the frequency domain. A better quantitative understanding of the phenomena of HHG was provided by the highly successful semi-analytical quantum theory of Lewenstein[8], where the electron is described quantum mechanically and the driving and emitted fields are still described classically. Many advances in the theory have since followed, particularly concerning accurate ab initio treatment of the HHG process from atoms and molecules[9–12], as well as the description of various HHG mechanisms from solids[13–16].

All such theories use quantum mechanics to describe the dynamics of the electrons in the driving field, however, the radiation emission and the driving field are treated classically. The emitted field is modeled as dipole radiation, with the dipole source calculated as the expectation value of the dipole moment of each driven atom[7,17]. Such an approach is semi-classical, which is not completely satisfactory and provides a consistent result with full quantum-electrodynamical calculations only in limited cases as was shown in[18]. The first theoretical approaches to quantize the fields that interact with the electron were built for a free electron in a monochromatic electromagnetic field[19–21]. Early pioneering theoretical studies developed a quantum electrodynamical formalism to describe HHG by quantizing the emitted field yet keeping the driving field classical[22,23]. More recent studies[24–26] developed a quantum formalism for HHG in which the electron states are dressed by the driving classical field, and the radiation is seen as spontaneous emission from these time-dependent dressed states. The driving field was also quantized in recent experimental and theoretical studies[27–32].

Nevertheless, to date, the conditions under which the spectral and statistical properties of the radiation differ significantly from the known effects of HHG seen in experiments remain unknown. It remains an open question whether quantum optics can produce new effects of intrinsically quantum nature, such as non-classical photon statistics or entanglement of the emission and emitting media. In particular, it is still undetermined whether the emitted HHG should be considered as an ensemble of photons, each with a single frequency (in a mixed state), or whether each photon is a quantum superposition of all frequencies in the comb. Answers to these questions would reveal new aspects of HHG with implications for attoscience, quantum optics, quantum electrodynamics (QED), and quantum information.

Here, we analytically develop a fully quantum theory of extreme nonlinear optics and use it to explore the quantum-optical nature of HHG. Our formalism does not assume a specific electronic system: it applies to atoms, molecules, or solids. We use the term "atom" in the sense of a general system. We present predictions for HHG in both the single-atom and the many-atom (i.e., an ensemble of atoms) regime, and we highlight in each case the deviation from the conventional treatments. In particular, in the single-atom case, we show that the spectrum would contain multiple shifted combs of HHG, which arise because of transitions between the initial and different final states of the driven

atom. We describe the transition between the single-atom and the many-atom regime and under which conditions it happens. We calculate the photon statistics of HHG and show that each spectral component can deviate from Poissonian statistics, showing squeezing and bunching, even when the emission is produced by many atoms, in contrast with anti-bunching in the single-atom regime. For both a single-atom and many-atom regimes, we find new features in the HHG spectrum arising from the breakdown of the dipole approximation to the emitted photons, which manifests in an emission of even harmonics (even from a monochromatic driving field). Most importantly, we show that each HHG photon is a comb with attosecond timescales and carries the entire spectrum's spectral content, which can be measured by a field autocorrelation experiment. Consequently, even a single photon carries information about the HHG process, including the energy distribution and the cut-off frequency, up until its observation.

## Results

**Quantum theory of extreme nonlinear optics and high harmonic generation.** In this section, we develop a general fully quantum framework for predicting the emission from an electronic system in a strong time-dependent external electromagnetic field. The formalism constructed here is based on the idea of using quantum electrodynamical perturbation theory for bound electrons that are dressed by an external field (SFQED), similar to the approach described in[24,26]. We go beyond the previous literature by considering fully quantized electromagnetic fields (for both the driving and emitted fields) and also taking into account beyond-dipole corrections. The quantization of electromagnetic fields enables analyzing the quantum statistics of the emitted field and the driving field.

We consider an electronic system driven by a strong laser field, which is described by a multimode coherent state: $|\psi_{\mathrm{laser}}\rangle = \prod_{\mathbf{k}\sigma} |\alpha_{\mathbf{k}\sigma}\rangle e^{-i\omega_{\mathbf{k}}t}$, where $\alpha_{\mathbf{k}\sigma}$ represents the coherent states' parameters that can be shown to be equal to the complex amplitudes of the Fourier components of the classical description of the incident driving field; $\mathbf{k}$ refers to the wavevector of a plane wave in free space, $\sigma$ to its polarization, and $\omega_{\mathbf{k}} = ck = c|\mathbf{k}|$ to its frequency, with $c$ the speed of light in vacuum. The combined wavefunction of the electronic system and the electromagnetic field, $|\psi(t)\rangle$ is determined by the Schrodinger equation

$$i\hbar \frac{\partial}{\partial t} |\Psi(t)\rangle = H|\Psi(t)\rangle, \tag{1}$$

where the (QED) Hamiltonian in the case of one electron is $H = \frac{1}{2m}(\mathbf{p} - q\mathbf{A})^2 + U + H_{\mathrm{F}}$, with $q$ being the electron charge, $m$ the electron mass, $U$ the atomic potential, and $H_{\mathrm{F}}$ the Hamiltonian of the free electromagnetic field. The quantized vector potential $\mathbf{A}$ contains both the driving field and the emitted field. This Hamiltonian is cited as a particular example, and the entire formalism below can be applied to any Hamiltonian. The solution of Eq. (1) for the combined wavefunction $|\psi(t)\rangle$ relies on three important steps. In the first step, we perform a unitary transformation on the Hamiltonian, which decomposes the vector potential $\mathbf{A}$ into a sum of a classical time-dependent part $\mathbf{A}_c(t) = \langle \psi_{\mathrm{laser}}(t)|\mathbf{A}|\psi_{\mathrm{laser}}(t)\rangle$ and a small quantum correction $\mathbf{A}_q = \sum_{\mathbf{k}\sigma} \sqrt{\frac{\hbar}{2\varepsilon_0 Vck}}[\mathbf{e}_\sigma a_{\mathbf{k}\sigma} e^{i\mathbf{k}\cdot\mathbf{r}} + \mathbf{e}_\sigma^* a_{\mathbf{k}\sigma}^\dagger e^{-i\mathbf{k}\cdot\mathbf{r}}]$, as shown in the Supplementary Note 1. Here, $V$ is a normalization volume, $\Sigma_{\mathbf{k}\sigma}$ is a summation over all photonic modes with polarization $\sigma$ and wavevector $\mathbf{k}$, the operators $a_{\mathbf{k}\sigma}$ and $a_{\mathbf{k}\sigma}^\dagger$ are annihilation and creation operators respectively, $\mathbf{e}_\sigma$ is a unit vector of polarization,

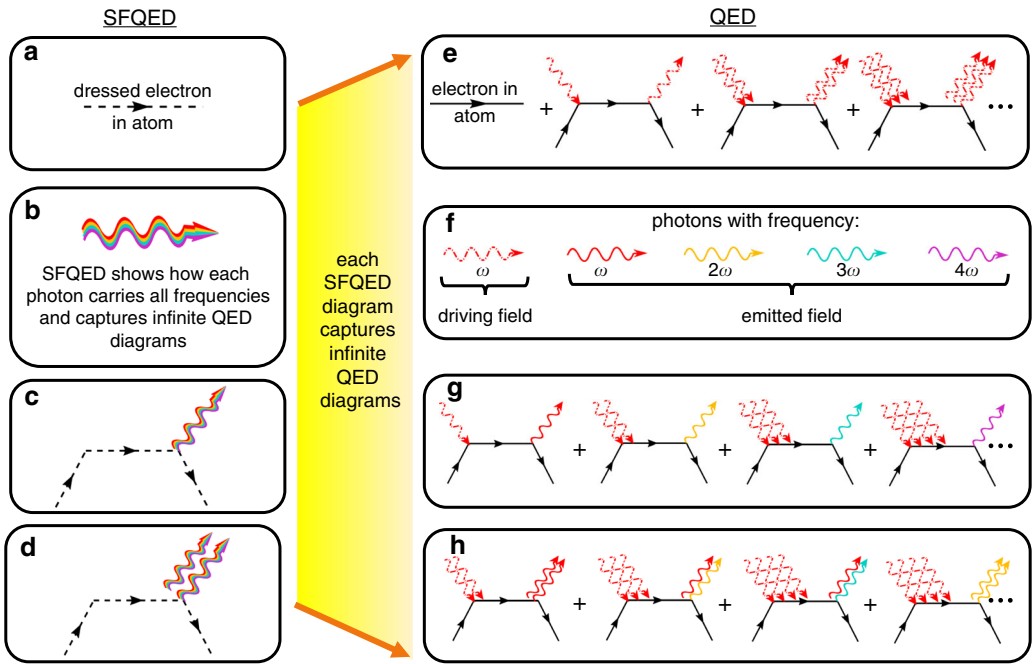

**Fig. 1 Diagrams in strong-field quantum electrodynamics (SFQED) versus ordinary quantum electrodynamics (QED).** In SFQED (**a**–**d**), we have perturbation theory in the weak emitted field: **c**, **d** correspond to different numbers of emitted photons. Each perturbation order in SFQED is equivalent to infinite diagrams of emission and absorption processes in the standard formulation of QED (**e**–**h**); **a**, **b** corresponds to zero-order SFQED, containing all the QED diagrams **e**, **f** necessary to describe the evolution of the wavefunction according to the time-dependent Schrodinger equation (Eq. 2); **c** corresponds to first-order SFQED (equivalent to **g** in QED), which we use to capture the effect of high harmonic generation; **d** corresponds to second-order SFQED (equivalent to **h** in QED), which can capture new processes in extreme nonlinear optics, such as laser-driven two-photon spontaneous emission.

and $\varepsilon_0$ is the vacuum permittivity. $\mathbf{A}_q$ and $\mathbf{A}_c$ describe the quantum emitted field and the classical driving field, respectively.

In the second step, we take advantage of existing analytical and numerical techniques that have been widely developed to solve the time-dependent Schrodinger equation (TDSE)[17,33,34]

$$i\hbar \frac{\partial |\phi_i(t)\rangle}{\partial t} = H_{\mathrm{TDSE}} |\phi_i(t)\rangle, \qquad (2)$$

where $|\phi_i(t)\rangle$ is the wavefunction of the electronic system, which initially (at $t = -\infty$) typically occupies the ground state or another eigenstate of the electronic system, but can also be in a superposition of eigenstates. The TDSE Hamiltonian, $H_{\mathrm{TDSE}}$, depends on the electronic system and for a single electron takes the form $H_{\mathrm{TDSE}} = \frac{1}{2m}(\mathbf{p} - q\mathbf{A}_c(t))^2 + U + H_{\mathrm{F}}$.

In the third step, we calculate the quantized radiation emission, described by the interaction of $\mathbf{A}_q$ with the strongly driven electronic state $|\phi_i(t)\rangle$ found in the second step. The coupling between an electronic system and the emitted field in typical nonlinear optics effects, such as HHG, is weak (Supplementary Note 5) and therefore can be accounted for by perturbation theory. The combined wavefunction $|\psi(t)\rangle$ of the electronic system and the photons is constructed from two parts: (1) the time-dependent electronic wavefunction $|\phi_i(t)\rangle$ is calculated using the classical driving field $\mathbf{A}_c(t)$, and (2) the photonic emission is calculated to first order in the quantized field $\mathbf{A}_q$. Altogether, the above recipe enables calculating general first-order processes in SFQED. This situation is shown in the left column of Fig. 1.

In the framework of SFQED, frequency conversion in nonlinear optical effects, such as HHG, is equivalent to a process of spontaneous emission with transitions between the electronic states $|\phi_i(t)\rangle$ that are dressed by the laser field. In the most general sense, spontaneous emission is the process of creating a photon in a mode with zero photons $|0\rangle$. Therefore, many nonlinear optical effects, including HHG and second harmonic generation, can all

be described in terms of (coherent) spontaneous emission in SFQED. The time-dependent state $|\phi_i(t)\rangle|0\rangle$, with $|0\rangle$ being a state with no emitted photons makes a quantum mechanical transition into states of the form $|\phi_i(t)\rangle|1_{\mathbf{k}\sigma}\rangle$. The entire set of the time-dependent states, $\{|\phi_j(t)\rangle\}$, evolves in time according to Eq. (2) and forms a basis for the electronic system. The photonic state $|1_{\mathbf{k}\sigma}\rangle$ represents one photon with wavevector $\mathbf{k}$ and polarization $\sigma$.

In the rest of this section, we demonstrate the results of our formalism for a general nonlinear optical process; the resulting formulas can also be straightforwardly applied to HHG. We use the dipole approximation to show explicit analytical expressions for the combined wavefunction $|\psi(t)\rangle$ and for the emission spectrum. We describe corrections beyond the dipole approximation in the next sections; however, in this section and the next section we use the dipole approximation, and in this case $|\psi(t)\rangle$ is given by (see Supplementary Notes 2 and 3):

$$|\Psi(t)\rangle = |\phi_i(t)\rangle|0\rangle$$
$$+ \frac{1}{\hbar}\sum_j\sum_{\mathbf{k}\sigma} e^{-i\omega_k t}\sqrt{\frac{\hbar\omega_k}{2\varepsilon_0 V}}\Bigg[\int_{-\infty}^t \Big(\mathbf{d}_{ji}(\tau)\cdot\mathbf{e}_\sigma^*\Big)e^{i\omega_k\tau}d\tau\Bigg]|\phi_j(t)\rangle|\mathbf{k}\sigma\rangle,$$
$$(3)$$

where $\mathbf{d}_{ji}(\tau) = \langle\phi_j(\tau)|q\mathbf{r}|\phi_i(\tau)\rangle$ are dipole matrix elements that can be complex. The summation $\Sigma_j$ is performed over all possible final electronic states $|\phi_j(t)\rangle$, labeled by different quantum numbers $j$. Each final (time-dependent) electronic state corresponds to a photonic state that is a superposition of various single-photon states with wavevector $\mathbf{k}$ and polarization $\sigma$. The result for $|\psi(t)\rangle$ is independent of the choice of the basis $|\phi_j(t)\rangle$.

Importantly, the superposition state in Eq. (3) suggests that any emission process in nonlinear optics, in which the electronic system varies in time, does not emit photons with different fixed frequencies, but rather each photon can be a superposition of multiple frequencies. This conclusion applies directly to HHG,

showing that each photon carries the entire HHG spectrum, containing all the spectral information of the HHG pulse, e.g., a single photon comb. This is a key conclusion that is illuminated by our formalism (or a single photon that represents a broad continuous spectrum if HHG is driven by few-cycle pulses).

The most remarkable feature of the combined wavefunction in Eq. (3) is the *entanglement* of the photonic state and the electronic state, in the sense that we cannot decompose the wavefunction to a tensor product of photonic and electronic states. The entanglement implies that there remains a connection between the photon and the emitting atom after emission even in very strong fields. This result is significant from a fundamental point of view because entanglement between an atom and a field, or between multiple photons, is usually only found in a weak perturbative regime (e.g., spontaneous parametric down-conversion by the perturbative $\chi^{(2)}$). However, here we show that entanglement can exist even in strong non-perturbative fields, which further motivates the study of HHG with the full-quantum description. Thus, from Eq. (3) we conclude that the wavefunction $|\psi(t)\rangle$ not only shows that each photonic state is a superposition of multiple frequencies but also shows that it has quantum features such as entanglement and carries the information about all transition matrix elements $\mathbf{d}_{ji}$. Our SFQED formalism also yields the emitted photon energy per unit frequency $\frac{d\varepsilon}{d\omega}$ (i.e., the spectrum), which follows immediately from modulo-squaring the final-state amplitudes of Eq. (3) and integrating over all photon emission angles and polarizations (see Supplementary Notes 2, 3). The spectrum is given by

$$\frac{d\varepsilon}{d\omega} = \sum_j \frac{\omega^4}{6\pi^2\varepsilon_0 c^3}\left|\int_{-\infty}^{+\infty}\mathbf{d}_{ji}(t)e^{i\omega t}dt\right|^2. \tag{4}$$

We can find $\mathbf{d}_{ji}(t)$ by solving the TDSE for each $j$, which can be done using any of the previously developed approaches[17,33,34]. We note that several of the qualitative conclusions we draw do not depend on the precise method used to obtain $\mathbf{d}_{ji}(t)$. This way, all the previous techniques developed for HHG (e.g., fully numerical approaches, strong-field approximation, etc.) can be facilitated for studying quantum effects. When solving the TDSE for a single electron in a single atom, Eq. (4) yields the general result for the spectrum of HHG from a single atom. Numerical results using Eq. (4) are presented in the next section.

**Conceptual differences between single-atom and many-atom high harmonic generation**. In this section, we show the numerical calculation of the HHG spectrum from a single atom in the 1D model of a helium atom. To emphasize the differences between single-atom and many-atom HHG, we compare our general result in Eq. (4) with the conventional formula of HHG[7,17,35], which can be derived from the general Eq. (4) (Supplementary Note 10) by neglecting contributions that arise due to transitions to different electron states $j$ different than i

$$\frac{d\varepsilon}{d\omega} = \frac{\omega^4}{6\pi^2\varepsilon_0 c^3}\left|\int_{-\infty}^{+\infty}\mathbf{d}(t)e^{i\omega t}dt\right|^2, \tag{5}$$

where $\mathbf{d}(t) = \mathbf{d}_{ii}(t) = \langle\phi_i(t)|q\mathbf{r}|\phi_i(t)\rangle$. Unlike Eq. (5) that contains only the expectation value of the dipole moment, our Eq. (4) contains all transition matrix elements $\mathbf{d}_{ji}$. We can always choose the basis of time-dependent states $|\phi_j(t)\rangle$ such that it includes $|\phi_i(t)\rangle$ and then the sum in Eq. (4) separates between one term $\mathbf{d}_{ii}(t)$ for the conventional result, and all the rest of the terms for the quantum corrections.

To observe the conceptual differences between Eqs. 4, 5, we performed a numerical calculation for a model of a helium atom, as depicted in Fig. 2. We studied the dynamics of a helium atom

(within the single active electron approximation) interacting with an external electric field with frequency $\omega$: $\mathbf{A}_c(t) = \frac{1}{\omega}\mathbf{E}_0\cos\omega t$. The initial state of the atom was chosen as its ground state $|\phi_1\rangle$, mimicking a 1s state (depicted in Fig. 2a). We calculated numerically the emission spectrum using Eqs. 4, 5. As shown in Fig. 2b, Eq. (4) yields much larger emission rates than Eq. (5). The differences in the spectrums of Eqs. 4, 5 are because transition matrix elements $\mathbf{d}_{ji}$ can be comparable to the element $\mathbf{d}_{ii}$ and even much larger, as depicted in Fig. 2c–f. Moreover, in Fig. 2b we can see that Eq. (5) yields a standard HHG spectrum with odd-only harmonics, while Eq. (4) yields no distinguishable discrete harmonic peaks. The lack of discrete peaks is a consequence of the low ionization potential of the excited dressed states, which ionize quickly and break the time-translation symmetry of the particular transition amplitudes[17,36].

Part of the features we presented in Fig. 2 can be corroborated by comparing with earlier work. Specifically, quantum corrections arising from transition elements $\mathbf{d}_{ji}$ were studied for a single xenon-like atom in[26,37]. There is an increase of background radiation (emission not at the integer harmonics) especially around low harmonics, which was predicted in[37] as well as in our study. This effect was shown to significantly change the spectrum for strong fields for which the cut-off is approximately at the 20th harmonic[37]. Our work shows that the quantum corrections can completely dominate the spectrum of emission for stronger fields for which the cut-off is approximately at the 60th harmonic. Beyond this comparison, each work emphasizes additional features, and uses a different formalism. We compare the two approaches quantitatively and prove mathematical connections in Supplementary Note 11.

We now discuss the emission from many ($N$) atoms. The emission of an ensemble of atoms can always be separated into a coherent and incoherent part. Particularly, the incoherent part of the emitted intensity is proportional to $N$, while the coherent part is proportional to $N^2$. The photon statistics in each case strongly depends on how much the atomic ground state changes during the HHG process. In cases for which the ground state only weakly changes[8] (related to the nondepleted ground state approximation), the coherent parts of the emission only arise from the $\mathbf{d}_{ii}$ elements and the emission is effectively classical, being governed by Poissonian statistics (Supplementary Note 5) and a factorizable (nonentangled) atom-field wavefunction. At the same time, the incoherent parts of the emission that arise from the $\mathbf{d}_{ji}$ elements contain features of entanglement and non-Poissonian statistics of light. In cases for which transitions between different time-dependent states $|\phi_j(t)\rangle$ are not negligible (i.e., breaking the nondepleted ground state approximation), we find nontrivial photon statistics for both the coherent and the incoherent parts of the emission, as we discuss in the next section.

The effects of incoherent contributions appear in many areas of physics and have been investigated for several decades (see e.g.,[38–41]). For a large enough ensemble, the incoherent part of the emission becomes negligible compared with the coherent part. We provide a qualitative argument to expose the coherent and incoherent parts of the radiation. Let us consider $N$ noninteracting atoms in a small volume, which interact with an external driving laser field, neglecting other interactions for the duration of the driving field (Supplementary Note 10). In the conventional case, if all the atoms were initially in the ground state, the contributions of the $\mathbf{d}_{ii}$ element of Eq. (4) from different atoms add up coherently, and the summed intensity is proportional to $N^2$. In contrast, contributions of all the other elements (involving states distinct from the initial state) from different atoms add up incoherently and the sum is proportional to $N$. For large values of $N$, we can eventually neglect the incoherent parts, and then, the many-atom HHG is adequately captured by Eq. (5) multiplied by $N^2$, in exact

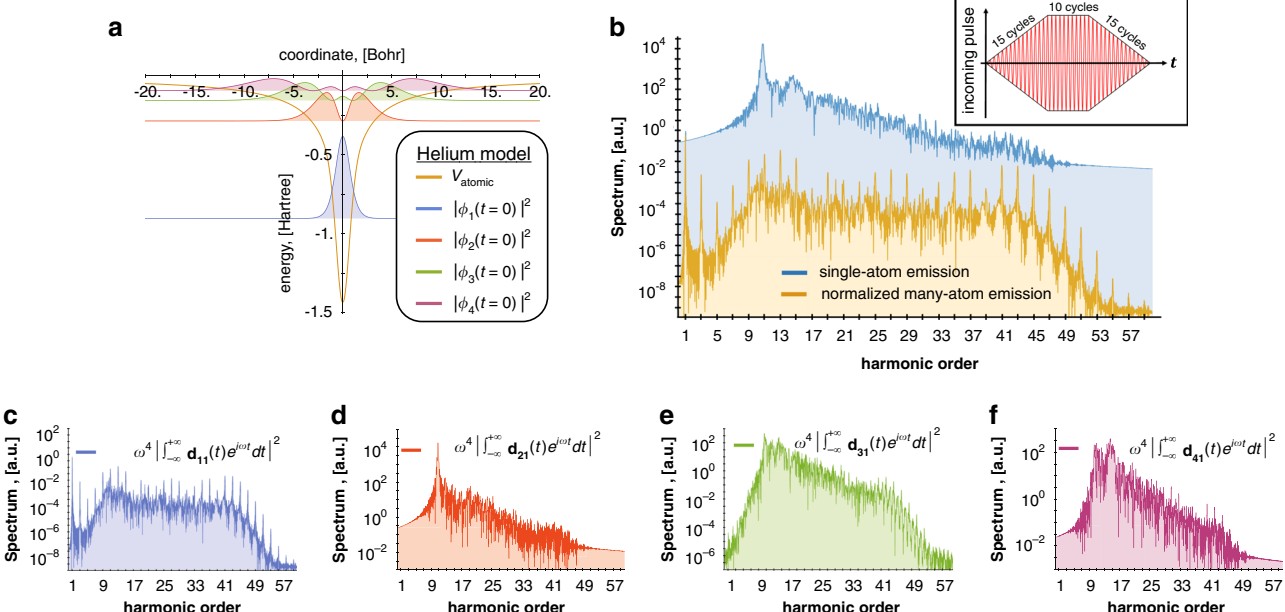

**Fig. 2 Single-atom and many-atom high harmonic generation (HHG). a** 1D model of a helium atom and its electronic states ($|\phi_1\rangle$, $|\phi_2\rangle$, $|\phi_3\rangle$, etc.) without an external driving field. This model of a helium atom is used for all the simulations in our work. **b** Emission spectrum calculated using Eq. (5) (yellow curve), which presents the HHG emission by many ($N \gg 10^4$) atoms normalized by $N^2$ vs. summing the first four elements in Eq. (4) (blue curve), which presents the emission by a single atom. **c–f** Contributions to the single-atom HHG spectrum, corresponding to different transition matrix elements, all having the same normalization. The wavelength of the driving field is $\lambda_0 = 800$ nm; the intensity of the driving field is $I = 2\,10^{14}$ W cm$^{-2}$. The form of the pulse is shown in **b**—trapezoidal pulse with 15 cycles rising on each side of trapeze and 10 cycles of the plateau of the trapeze. In many-atom emission, only **c** gives a coherent contribution ($\sim N^2$) to the emission, while **d–f** give an incoherent contribution ($\sim N$).

agreement with the conventional classical theory[2,6,7,42]. We derive quantitatively this in the Supplementary Note 10.

Nevertheless, there exist conditions under which the incoherent part of HHG emission becomes significant even for a large ensemble. Notably, in our numerical example in Fig. 2, the magnitude of $|\mathbf{d}_{21}|^2$ (the largest matrix element) is $10^4$ times greater than the magnitude of $|\mathbf{d}_{11}|^2$ (Fig. 2). Hence, when the number of active atoms is $N < 10^4$, we expect significant observable deviations from the spectrum of the conventional HHG theory Eq. (5), i.e., "quantum corrections". This scenario contains features that cannot be explained by the semi-classical approach, and it thus underscores the importance of the SFQED approach and shows perspectives for the creation of macroscopic (at least mesoscopic) quantum states of entangled light and matter.

**Photon statistics of high harmonic generation**. In this section, we investigate the photon statistics of each harmonic in HHG. Specifically, we numerically calculate the Mandel parameter $Q$ and the squeezing $\eta$ of the emitted light at each frequency. The Mandel parameter is defined as $Q = \frac{\langle n^2 \rangle - \langle n \rangle^2}{\langle n \rangle} - 1$, where $n$ is the photon number operator[43]. In addition to the Mandel parameter $Q$, which is a convenient way to summarize the quantum statistics of the photons, we also consider the squeezing factor $\eta = 10|\log(4\Delta X^2)|$ of the emitted light, where $\Delta X^2$ is the variance of the "position" quadrature operator, related to the vector potential[43]. For the single-atom regime, the squeezing is negligible; however, for the many-atom regime squeezing becomes significant. Interestingly, the photon statistics in the single-atom regime and many-atom regimes differ significantly. We present these two different regimes separately in Figs. 3, 4. Additional information about the formalism, approximations, and numerical calculations for both cases is provided in Supplementary Note 6.

The emission in the single-atom regime is very weak and thus the squeezing $\eta$ of the emitted light is negligible and the Mandel parameter $Q$ is very small. Nevertheless, the emission has nontrivial quantum properties, showing *super-Poissonian* and *sub-Poissonian* statistics simultaneously for different spectral components (red and blue colors in Fig. 3a, respectively). We find that the super-Poissonian statistics for the lower harmonics (first ten in Fig. 3a) is connected with transitions inside the atom that end up in the ground state (e.g., $|\phi_i\rangle \rightarrow |\phi_{j\neq i}\rangle \rightarrow |\phi_i\rangle$). Meanwhile, we find that the transitions $|\phi_i\rangle \rightarrow |\phi_{j\neq i}\rangle$ lead to sub-Poissonian statistics at the higher harmonics (beyond ten in Fig. 3a). These transitions have a negligible contribution for lower harmonics.

Figure 3b elaborates on this trend, showing that the Mandel parameter $Q$ has a sharp negative-valued peak (strongly sub-Poissonian) *at exactly* the energy of the transition between the ground and the first excited state (Fig. 3c). For higher harmonics, the transitions involve ionization (transition in the continuum), which causes the Mandel parameter $Q$ to decay (elaborated in Supplementary Note 6). Altogether, although the number of photons emitted from HHG in the single-atom regime is generally small, the emission has nontrivial statistics that strongly depends on the spectrum of the atom (note that when $n$ is small, $Q$ can be very small and still represent light with strongly nonPoissonian statistics).

The emission in the many-atom regime, surprisingly, also has quantum features: super-Poissonian photon statistics for all the frequencies. The reason is similar to the low harmonic emission in the single-atom regime. In the many-atom regime, for all harmonics, the dynamics are dominated by transitions inside the atom for which the final state is the ground state (e.g., $|\phi_i\rangle \rightarrow |\phi_{j\neq i}\rangle \rightarrow |\phi_i\rangle$), leading to super-Poissonian statistics. Moreover, in the many-atom regime, there can be significant

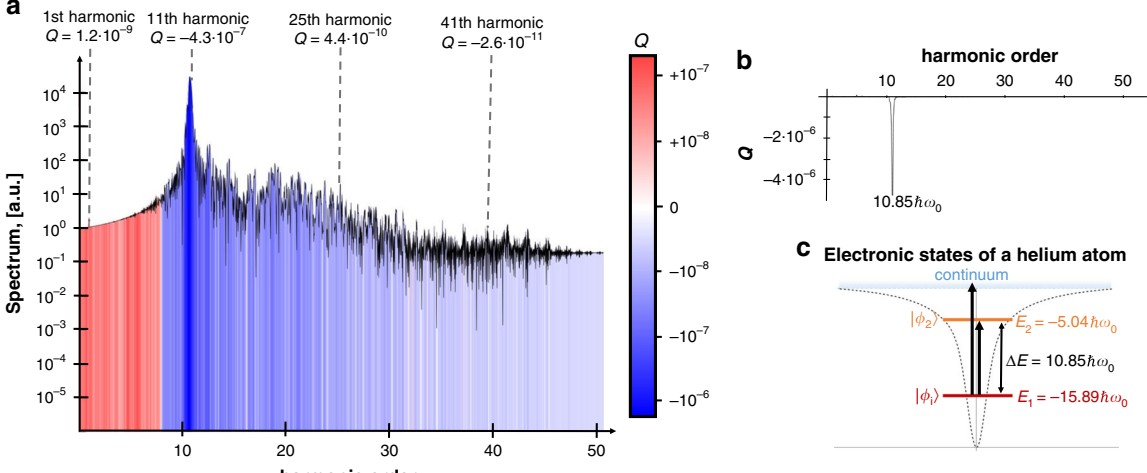

**Fig. 3 Photon statistics of HHG in the single-atom regime. a** HHG spectrum with the background color showing the Mandel parameter $Q$; red corresponds to super-Poissonian statistics ($Q > 0$), white corresponds to Poissonian statistics ($Q = 0$), and blue corresponds to sub-Poissonian statistics ($Q < 0$). **b** Dependence of the Mandel parameter $Q$ on the frequency of the emitted radiation. **c** Illustration of the electronic states of the helium atom. The energy difference between ground and excited level equals to the frequency of resonance in the Mandel parameter. All the parameters of the numerical calculations are the same as in Fig. 2. Our numerical calculations assume a single atom, although the same conclusions apply up to several thousands of atoms with a larger $Q$ (Supplementary Note 6).

squeezing. Both the value of the Mandel parameter $Q$ and the squeezing $\eta$ increase with the number of phase-matched atoms $N_P$ (Fig. 4). Note that the number of phase-matched atoms can be many times smaller than the total number of atoms, i.e., $N_P \ll N$. To get more squeezing of light, we need to have a larger density of atoms, e.g., the larger density of the gas, in the case of atomic HHG. Figure 4 shows how the Mandel parameter $Q$ and the squeezing $\eta$ depend on the number of phase-matched atoms $N_P$ and give quantitative estimates of these parameters in the many-atom regime.

To conclude this section, we investigated the photon statistics of HHG for both the single-atom and many-atom regimes and found that both can deviate from Poissonian statistics in different ways. The predictions of super-Poissonian statistics of HHG in the many-atom regime can be readily observed using conventional parameters in current HHG experiments. It is a question of fundamental interest to also find avenues to observe features of the single-atom regime, where SFQED could enhance the normally-incoherent parts of HHG, and reach new effects of many-body systems with nontrivial photon statistics. A full investigation of quantum many-body effects of SFQED is left for future work.

**Effects in high harmonic generation beyond the dipole approximation.** In all previous work on HHG, the dipole approximation was used for calculating the emission. A few papers and books, e.g.,[17,44–46], considered beyond-dipole effects for the driving field but not for the emitted field. In Eqs. 2–5 and the numerical simulations of Fig. 2, we applied the dipole approximation to both the driving field and the emitted field. In this section, we explore the effects of breaking the dipole approximation for the emitted field. In other words, we demonstrate the corrections to the HHG spectrum that result from the extremely short wavelength of the emitted photons themselves (on the same order of or smaller than the size of the electron wavefunction in the driven system).

First, we give an analytical estimate of the conditions in which the dipole approximation can be broken. We estimate that the effective size of the electron wavefunction during the interaction with the strong driving laser is on the order of magnitude of the

quiver radius $a = \frac{q\lambda_0^2}{4\pi^2 mc^2}\sqrt{\frac{2I}{c\varepsilon_0}}$, where $\lambda_0$ is the wavelength of the driving field. Since $a$ is much smaller than $\lambda_0$, the dipole approximation is accurate for the *driving field*. A typical ratio is $\frac{2\pi}{\lambda_0}a \sim 10^{-2}$ for $\lambda_0 = 800$ nm and $a = 1$ nm. The use of a plasmonic environment for confining the field can in principle break the dipole approximation for the driving field, as was previously proposed for HHG[44,45,47] and other effects (see e.g., ref. [48]), and yet, plasmonic enhancements of HHG do not currently show evidence of such corrections[49,50].

We find that the *emitted field* can break the dipole approximation in realistic conditions, which can have subtle implications. The dipole approximation for the emitted field becomes gradually less accurate as the harmonic number increases, since the emission wavelength $\lambda$ can reach the single nanometer scale for harmonics $n$ of several hundred (even more was observed[51]). We want to clarify the difference between the two different types of beyond-dipole corrections, which are fundamentally different and are related only by name. Beyond-dipole corrections due to the *driving field* change the TDSE and the resulting wavefunction $|\phi(t)\rangle$, but do not change the form of Eq. (5). Such corrections to the evolution of the wavefunction include relativistic corrections to the Schrodinger equation (see e.g., ref. [42,52]). In contrast, the beyond-dipole corrections to the *emitted field* described in this work do not change the wavefunction in the matrix element but modify the form of Eq. (5), as we see below.

The HHG emission power scales with the dimensionless parameter $x = \frac{2\pi}{\lambda}a$, or equivalently $x = n \cdot \frac{2\pi}{\lambda_0}a$, and thus, when $x$ approaches unity we predict that multipolar corrections can become significant. Scenarios in which the emitted field breaks the dipole approximation have been neither previously observed nor proposed in the context of HHG. Related effects were previously predicted for emission into modes of confined light (e.g., in polaritons in 2D materials[53]) and shown when the emitting electron wavefunction is significantly enlarged (e.g., using Rydberg states[54]). Here, we predict that HHG can break the dipole approximation even in the case of regular atoms emitting into free-space radiation, provided sufficiently high harmonics: e.g., a few hundred[55,56] cause $x$ to approach unity. Considerably

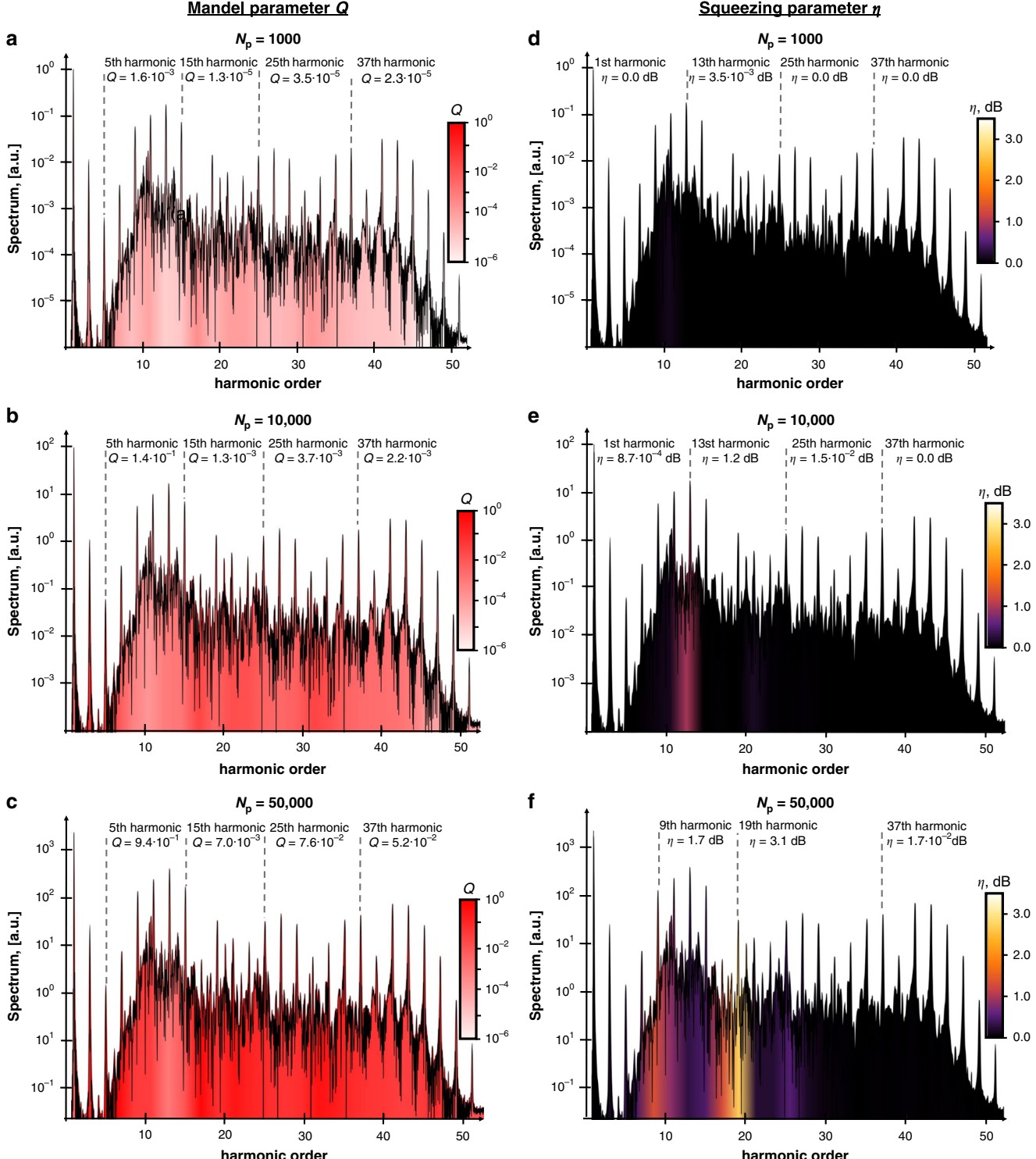

**Fig. 4 Photon statistics of HHG in the many-atom regime. a–c** present the Mandel parameter $Q$ by the intensity of the red color for different numbers of phase-matched atoms $N_p = 10^3$, $10^4$, and $5 \times 10^4$ respectively. **d–f** present the squeezing $\eta$ for the same numbers of phase-matched atoms. With an increase in the number of phase-matched atoms, the quantum properties of the emitted light become larger. All the parameters of the numerical calculations are the same as in Fig. 2.

higher harmonics have been observed[51], and thus, by fulfilling the conditions described below, we expect our predictions to be readily observable in existing HHG setups.

To quantify the implications of breaking the dipole approximation, we start from the Schrödinger equation (Eq. (1)) and Hamiltonian without the dipole approximation for the emitted field

$H = \frac{1}{2m}\left(\mathbf{p} - \frac{q}{c}\mathbf{A}\right)^2 + U + H_{\mathrm{F}}$. In the many-atom regime, first-order perturbation theory in the weak emitted field leads to the spectrum of emitted energy per unit solid angle $\mathrm{d}\Omega$ per unit frequency $\mathrm{d}\omega$:

$$\frac{\mathrm{d}\varepsilon}{\mathrm{d}\omega\mathrm{d}\Omega} = \frac{\omega^2 q^2}{16\pi^3\varepsilon_0 m^2 c^3}\sum_\sigma\left|\int_{-\infty}^{+\infty}\left(\mathbf{P}_{\mathrm{ii}}(t)\cdot\mathbf{e}_\sigma^*\right)e^{i\omega t}\mathrm{d}t\right|^2, \quad (6)$$

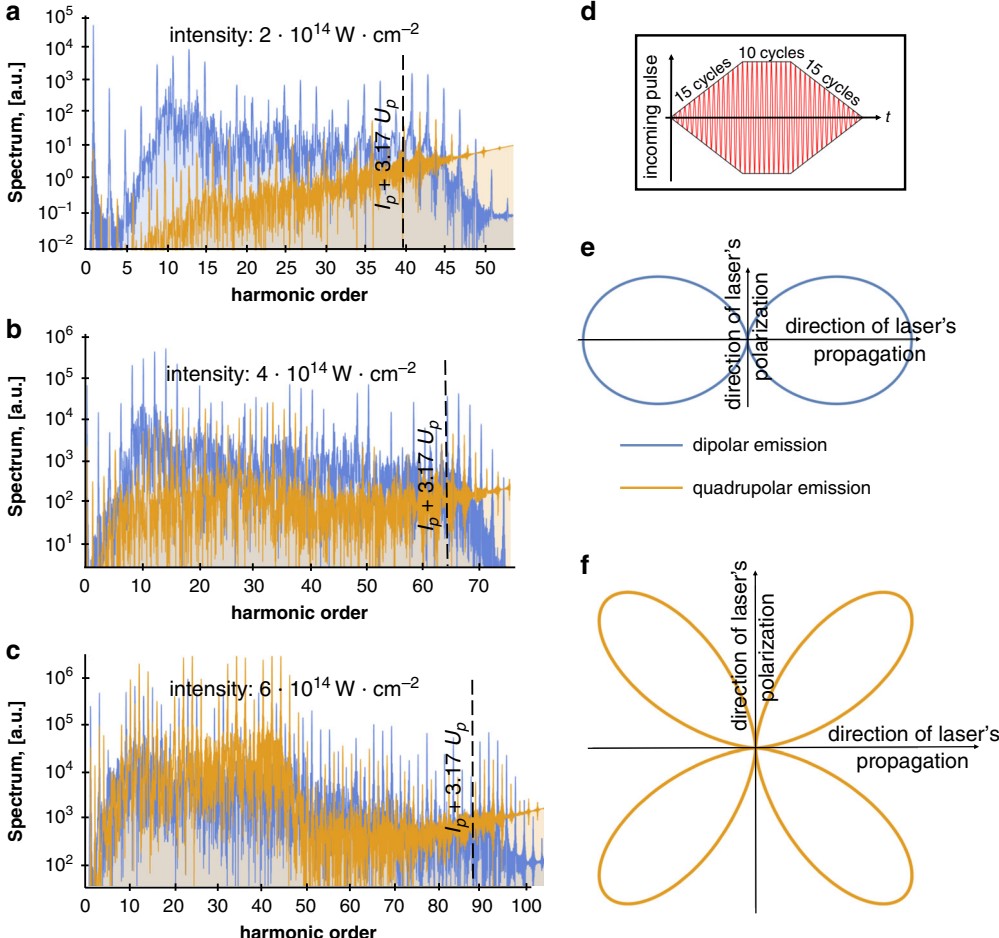

**Fig. 5 Breakdown of the dipole approximation for the emitted high harmonic radiation.** Numerical calculations of the spectrum use a 1D model of a helium atom. **a–c** Spectrum of dipolar (blue curve) and quadrupolar emission (yellow curve) for different intensities according to Eq. (7). The dashed vertical lines mark the conventional HHG cut-off energy $I_p + 3.17U_p$, where $I_p$ is the helium ionization potential and $U_p$ the ponderomotive energy. The quadrupolar emission gives a significant contribution to the even harmonics and becomes larger for larger intensities. The wavelength of the driving field is $\lambda_0 = 800$ nm; the intensities of the driving field are $2 \times 10^{14}$ W cm$^2$, $4 \times 10^{14}$ W cm$^{-2}$, and $6 \times 10^{14}$ W cm$^{-2}$, respectively. **d** The form of the incident pulse-trapezoidal pulse with 15 cycles rising on each side of trapeze and 10 cycles of the plateau of the trapeze. **e, f** Angular dependence of dipolar (blue) and quadrupolar (yellow) emission, respectively. The angular dependence in the insets is derived analytically (see Supplementary Note 9). The Ox and Oy axes are the directions of propagation and the direction of polarization of the driving field, respectively.

where $\mathbf{P}_{ii}(t) = \langle \phi_i(t) | e^{-i\mathbf{k}\cdot\mathbf{r}} \left( \mathbf{p} - \frac{q}{c} \mathbf{A}_c(t) \right) | \phi_i(t) \rangle$ (Supplementary Note 2). We expand Eq. (6) in multipoles to third-order (Supplementary Note 9). After the integration over $d\Omega$ we get the spectrum:

$$
\frac{d\varepsilon}{d\omega} = \frac{\omega^4}{6\pi^2 \varepsilon_0 c^3} |\langle \mathbf{d}(\omega) \rangle|^2
$$
$$
+ \frac{\omega^4}{6\pi^2 \varepsilon_0 c^3} \frac{\omega^2}{5q^2 c^2} \left( \frac{1}{4} |\langle \mathbf{d}^2(\omega) \rangle|^2 - \frac{1}{3} \mathrm{Re} \left[ \langle \mathbf{d}(\omega) \rangle \cdot \langle \mathbf{d}^3(\omega) \rangle^* \right] \right),
$$
$$
(7)
$$

where $\langle \mathbf{d}(\omega) \rangle = \int_{-\infty}^{+\infty} \langle \phi_i(\tau) | q\mathbf{r} | \phi_i(\tau) \rangle e^{i\omega\tau} d\tau$, $\langle \mathbf{d}^2(\omega) \rangle = \int_{-\infty}^{+\infty} \langle \phi_i(\tau) | (q\mathbf{r})^2 | \phi_i(\tau) \rangle e^{i\omega\tau} d\tau$ and $\langle \mathbf{d}^3(\omega) \rangle^* = \int_{-\infty}^{+\infty} \langle \phi_i(\tau) | (q\mathbf{r})^3 | \phi_i(\tau) \rangle e^{-i\omega\tau} d\tau$. The first term in Eq. (7) gives the regular dipolar emission (the same as Eq. 5) and the second term gives the quadrupolar emission.

In Fig. 5, we calculate the relative contributions of dipolar (blue) and quadrupolar (yellow) emission to the HHG spectrum for different driver intensities. While the dipolar emission gives a larger contribution to odd harmonics, the quadrupolar emission gives a large contribution to even harmonics. Therefore, even

harmonics can be used to observe quadrupole corrections to HHG. The quadrupolar emission increases with the driving field intensity faster than the dipolar emission. Moreover, for high harmonics (e.g., soft x-ray photons), quadrupolar contributions become comparable to dipolar contributions. Of course, additional higher-order multipolar corrections can become important too, and at even higher harmonics and higher driver intensities, the multipolar expansion fails and require calculating Eq. (6) without approximations.

Dipolar and quadrupolar emissions also have a very different directionality, as can be seen in the Fig. 5d, e. Whereas dipolar emission is in the direction of the propagation of the driving laser, quadrupolar emission has zero intensity in the propagation direction. When the emission is from many atoms in a large area of interaction relative to the wavelength of the driving laser, the angular distribution of HHG also strongly depends on phase-matching[7,35] (akin to many other nonlinear processes). Phase-matching in HHG from a single driving laser pulse, in general, leads to a strongly directional emission in the direction of the driving field propagation. This condition will enhance the dipolar emission and inhibit the quadrupolar emission, which may explain why the breakdown of the dipole approximation has not

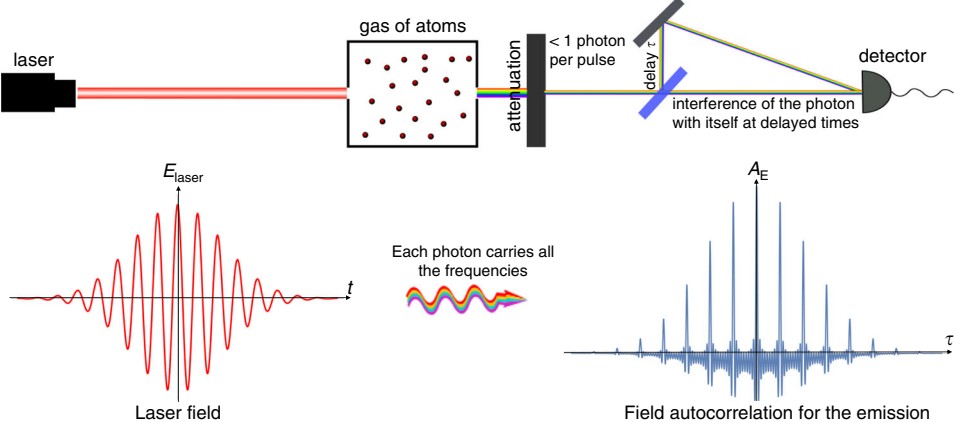

**Fig. 6 Proposed autocorrelation experiment: scheme to observe that each high harmonic generation (HHG) is a frequency comb.** HHG emission is first created in a typical setup, e.g., where the source is a gas of atoms. We attenuate the output emission and measure the autocorrelation, varying the delay, for different levels of attenuation. If each photon carries the entire spectral content of the HHG emission, then the normalized autocorrelation function would be independent of the number of photons per HHG pulse, even when it is less than one photon on average per pulse (see Supplementary Note 7).

been observed previously. However, in general, the emission direction can be manipulated in various ways. For example, the gas can be confined in a small volume or a thin layer comparable with the wavelength of the driving field (especially relevant for solid HHG[13–16,29]). Alternatively, the driving field can be a superposition of excitations from a few directions with different wave vectors, as described in[57,58]. In such cases, when the HHG emission is not unidirectional, effects beyond the dipole approximation could play a more important role and change significantly the physics of HHG.

**Each high harmonic generation photon is a comb: discussion of an experimental scheme.** The classical picture of HHG describes coherent multifrequency emission that can create an attosecond comb, and yet, the precise quantum optical nature of the emission is not clear from such a picture. The extremely nonlinear nature of the process raises many possible descriptions of the quantum nature of the emission, especially since it is constructed from many QED diagrams (Fig. 1, right column) in a highly nonperturbative manner. Without the guidance of a complete quantum picture, the emission was, for instance, considered to be made up of different photons of different frequencies or photons all having the same frequency in an entangled state with the driving laser. Below we discuss the consequences of our SFQED formalism on the level of each single photon in HHG.

Our formalism reveals the nature of the HHG emission: each emitted photon carries all the frequencies of the HHG process. To test this interpretation, we propose the following experiment. Beginning with a typical HHG source (e.g., a gas of atoms), we can attenuate the output emission to leave on average (less than) one photon per driving laser pulse (Fig. 6). We can then measure the autocorrelation function for the field passing through the attenuator. The autocorrelation result as a function of the attenuator strength could show the quantum optical nature of HHG when the intensity of the transmitted field is reduced from that of a classical field toward the single-photon limit (Fig. 6). Our formalism shows that normalized autocorrelation and other measurable quantities will remain the same for any number of photons (only up to the challenge of a lower signal-to-noise ratio) (see Supplementary Note 7). Such an experiment would lead to a different result for other photonic states and thus could distinguish between them. For example, if the emission is described by classical statistics, a normalized autocorrelation

function would depend on the number of output photons, and in the single-photon limit, it would have the form of a cosine function. This experimental setup is very close to that described in ref. [59]. Future work could explore other experimental proposals, such as higher-order autocorrelation functions, e.g., intensity autocorrelations, and optimize the conditions for such experiments to accommodate the inevitably low signal.

## Discussion

We developed a fully quantum formalism that captures general processes of extreme nonlinear optics, and we demonstrated it for the HHG process. We found new effects that arise from the quantum theory and cannot be described by the conventional theory—effects on the level of both the single HHG photon and the macroscopic photonic state. Our predictions include the emission of multiple spectral combs, the photon statistics of each spectral component, beyond-dipole effects in angle and frequency, and the exact structure of each single photon in HHG. Squeezing and nonPoissonian statistics have numerous applications in the field of sensing, high resolution imaging, weak measurements and quantum communications.

Our formalism can be straightforwardly generalized in many different directions. We can generalize our theory for systems of many electrons and for solids, which are of major interest for the research community[13–16,29]. It is also possible to find the next relativistic corrections to our nonrelativistic theory and take into account magnetic dipole effects[60]. Most generally, the theory we advance in this manuscript can be applied to all extreme nonlinear optical processes, and thus, we expect that our theory will guide the discovery of new quantum effects in other areas of nonlinear optics. For instance, the formalism is also capable of reproducing well-known perturbative light-matter interaction effects, such as spontaneous emission (see Supplementary Note 3). Special cases of our formalism describe the nonlinear Compton Effect[61,62] and multiphoton Thompson scattering[63].

Looking forward, one of the most important new aspects our formalism reveals is the *entanglement* between the photons and the emitting atoms, which remains after the emission even in very strong fields. This entanglement may have important consequences in the field of quantum optics and pump-probe experiments. For example, it could guide the development of entangled attosecond pulses in the EUV or soft-X-ray regime, which may have direct applications for metrology and precision imaging. The entanglement can also help in the development of novel heralded single-photon sources, such as single-photon frequency-combs.

From a fundamental standpoint, finding truly quantum effects in *ensembles* of atoms will open new perspectives for HHG and many other nonlinear optical effects. While a complete formulation of many-body HHG is still elusive, it may reveal promising opportunities to enable truly many-body quantum electrodynamical effects with a large number of photons, which will have important consequences from a fundamental point of view.

## Methods

**Numerical simulation of high harmonic generation.** The time-dependent Schrodinger equation for 1D helium atom (Eq. 2) was solved numerically using 3rd order split-step approach with absorbing boundary conditions.

## Data availability

The data supporting the findings of this study are available from the corresponding author upon reasonable request.

## Code availability

The code supporting the plots within this paper is available from the corresponding author upon reasonable request.

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

## Acknowledgements

This work was supported by the by the Israel Science Foundation (Grant No. 1781/18 and No. 831/19) and the Wolfson foundation. O.N. gratefully acknowledges the support of the Adams Fellowship Program of the Israel Academy of Sciences and Humanities. I. K. acknowledges the support of the Azrieli Faculty Fellowship. A.G. gratefully acknowledges the support of the Sherman fellowship.

## Author contributions

All authors made substantial contributions to all aspects of the work.

## Competing interests

The authors declare no competing interests.
