## [Peer Review File · Nature Communications]

Reviewers' Comments:

Reviewer #1:

Remarks to the Author:

The review on the manuscript

"On the Quantum-Optical Nature of High Harmonic Generation" by Professor Kaminer et al)

The authors of the paper propose an approach that allows one to analyze from the first principles the emission from a quantum system (atom, molecule, etc) in the presence of a strong laser field. Specifically, this approach is implemented for analyzing the high order harmonics generation (HHG) from a model atom.

The approach is based on the ab initio consideration of the interaction of a quantum-mechanical system (atom, molecule ...) with the modes of a quantized field. The intense laser field acting on the system principally can be treated also as quantum object. But from the very beginning of the paper laser field considered from the standpoint of classical theory. In this case, the interaction of an atom with an electromagnetic vacuum is considered in the first order of the perturbation theory (i.e., only single-photon excitations of field modes appear to exist), while the action of a strong laser field is accurately taken into account. I believe that the development of this approach is fundamentally important, because any radiation of a quantum system at frequencies that do not coincide with the frequency of the acting field always starts with spontaneous emission, i.e. it is the result of interaction with the electromagnetic vacuum surrounding the atom. I am not sure, that is possible to study the appearance of such frequencies in the radiation spectrum, since the semiclassical theory, which is commonly used to analyze the radiation emission from an atom at different frequencies (and in particular, the HHG), is not completely satisfactory (see, for example, *Laser Phys. Lett.*, V. No. 4, 045301 (2016)). I believe that a link to this work in the text of the paper would be more than appropriate. In this sense, the stated research topic is fundamentally important. Also important is the analysis of the radiation emission and HHG from an ensemble of atoms. Such analysis is especially important for practical applications as in real experiments the emission is observed from the atomic ensemble but not a single atom. The specific features of extremely high order harmonics emission are analyzed beyond the dipole approximation. The possibilities of experimental observation of the effects arising in the proposed approach that are principally cannot be understood within the frames of a semiclassical consideration of the problem, are discussed. Attention is drawn to the fact that in the process of interaction the system "atom + quantized field modes" turns out to be entangled, and, therefore its wave function cannot be factorized into the wave functions of the atomic and field subsystems.

However, there is a number of fundamental remarks on the paper material.

1) First of all, it should be noted that the approach formulated in the paper regarding the consideration of a single atom dressed by the classical field and interacting with an electromagnetic vacuum (this interaction is taken into account in the first order of the perturbation theory) is actually (in its physical essence) equivalent to the approach proposed in [20,21] from the list of references. I believe that this statement should be clearly stated in the paper under review. Otherwise, if the authors do not agree, it should be clearly stated what is the essential difference between the proposed approach and the approach described in (*EPL*, V.116, No. 1, P. 14003 (2016); *JETP*, V. 125, No. 4, P. 587-596 (2017)). The second of these papers should also be cited. In this sense, the novelty of the reviewed paper is the possibility of analyzing of the laser field also in the frames of quantum electrodynamics. This can be important if, for example, the atomic dynamics in "squeezed" electromagnetic fields is analyzed. However, the authors only declared but did not consider in detail such quantum approach to the description of the laser field. I would also mention that the typo in the

surname of one of the authors [20,21] exists.

2) The authors focused on the analysis of the spectrum of spontaneous emission (HHG) arising from the interaction of the Ti-Sa laser radiation with a model one-dimensional helium atom. However, a similar problem was already considered earlier in (Laser Phys. Lett., V.14, No. 5, 055301 (2017); JETP, V. 125, No. 4, P. 587-596 (2017)), but for a model one-dimensional xenon atom. It seems to me absolutely necessary, however, to compare the results of calculations performed in the present paper with those published earlier, at least at a qualitative level.

3) In addition to remark 2. Not only the frequency of the incident laser radiation (Ti-Sa laser) and its intensity (2×10^{14} W/cm²), but also the pulse duration and its envelope should be clearly indicated in the paper. Without such data, the data presented in the paper (see Fig. 2) and the corresponding explanations in the text seem to be incomplete.

4) The authors claim that they describe the dynamics of the atomic system in a classical field analytically. I would like to see how this is done? Is it the SFA approximation? And is it possible to do this for an arbitrary envelope of a laser pulse and its duration, in particular, under conditions of a substantial depletion of the ground (initial) state of an atom?

5) Undoubtedly, from a practical point of view, an important element of the paper is the transition to a description of the radiation from an ensemble of atoms. But this transition was made somehow unintelligible with references to rather old publications that are not directly related to the problem. In such a situation a quantum system has become more complex: it is already a number of emitting particles (this is equivalent simply to increasing of the dimension of the space in which the atomic wave function is given). And to describe the formation of a macroscopic radiation field, we must abandon the single-photon approximation for radiation field and consider the possibility of emitting a certain number of photons. Under definite conditions such an approach, in particular, would allow the authors to describe the effect of superradiance of atoms dressed by the field. The effect, which is essentially quantum in nature. It seems to me that the authors here should rework the text of the paper in this direction, and not only speculate enough about proportional terms N^2 and N .

To conclude, in my opinion the material presented is important and interesting, but is needed in significant revision.

Reviewer #2:

Remarks to the Author:

I recommend publication of "On the Quantum-Optical Nature of High Harmonic Generation," by Gorlach et al. This well-written manuscript highlights a program of analyzing laser high harmonic generation from the standpoint of QED. The authors couple a single-active-electron one-dimensional version of helium to the quantized electromagnetic field following a perturbative approach. They comment on a host of quantum effects that can be seen using their model. Each of these effects might be the subject of a separate publication. By commenting on all of them, the authors apparently feel that this work merits publication in Nature Photonics. I tend to agree. In this case, their overall approach to the problem is emphasized, illustrating the variety of effects that can be studied with it. I found the discussion surrounding Fig. 2 particularly interesting. The authors make it clear that, what researchers using the semiclassical approach often loosely call "single-atom" calculations, is really an ensemble average of atomic dipoles in a region of space small compared to a laser wavelength. They show how the true single-atom spectrum shows a continuum of wavelengths, whereas ensemble phase matching pulls out the discrete dipole spectrum.

The authors also find that individual photons should carry information about the entire high harmonic spectrum. They propose a basic photon correlation experiment to demonstrate this. To realize this experiment, it may be difficult to find an attenuator that transmits a wide spectrum uniformly. Still, the concept is interesting.

The authors comment on the failure of the dipole approximation for the very high-order harmonics, when the emission wavelength is no longer much larger than the trajectory path of the electron. In that case, the quadrupole term becomes important.

I have analyzed this manuscript as a person with a background in high harmonics, not as an expert in QED. I am unable to judge technical soundness of the QED calculations.

Reviewer #3:

Remarks to the Author:

The title and half of the abstract of the present manuscript are very exciting. One expects to see in the paper, as stated in the first half of the abstract, investigation of the statistical properties of the emitted HHG light and learn how and at which conditions a squeezed HHG light can be created.

Unfortunately, the text of the paper brings disappointment. The paper put forward three main results: 1) Each HHG photon is a spectral comb, 2) Incoherent HHG can be emitted when recombination occurs to the bound state different than the original state, 3) For HHG at very high energies, nondipole emission should be accounted for.

The HHG community is well aware of all three points. Moreover, the second and third points are not directly related to the quantization of the emitted photon field. On the first point, it is well accepted that the HHG emission is a coherent process and the emitted radiation field is a coherent field. The coherent harmonic field described in the quantum language is, in fact, equivalent to the collection of photons each of which is a spectral comb.

The second point not only well known but it is generally not used in applications, because it creates incoherent radiation which is minor if the medium is not very dilute. The authors' contribution is that they write a formal wave function in Eq.(3) and indicate on the entanglement between the atom and the photon field. However, the reader expects from the paper to learn about "intriguing consequences in the field of quantum optics" which emerge due to the entanglement. It is stated in the paper but not shown.

Finally, on the third point, the nondipole effects at a photon emission will be important at photon energies about 10 keV, to have the wavelength of the emitted photon comparable to the atomic size. This is already becoming achievable in experiments and nondipole effects are discussed in many publications, see e.g. M. Klaiber, et al., PRA 75, 063413 (2007), and a review [M. C. Kohler, et al. Frontiers of atomic high-harmonic generation Adv. Atom. Mol. Opti. Phys. 61, 159 (2012)], which includes many other citations on this topic.

Concluding, the present manuscript contains results of minor novelty and I cannot recommend it for Nature Communications. After significant revision, combining the supplement with the paper, providing detailed experimental setup for measuring the single HHG photon quantum properties, with an appropriate feasibility study, providing an example of application of the atom-photon entanglement, and elaborating more on the nondipole features in the HHG spectra, the paper may be suitable for publication in Scientific Reports.

Responses to referee #1 comments:

We thank the referee for taking the time to review our manuscript, and for providing us with helpful comments, which have improved the quality of our paper. We also wish to thank the referee for stating that our work is fundamentally important. Below we respond to the referee's comments point-by-point, where his/her comments are in **bold**, our answers denoted by ordinary text and changes in the manuscript are written in **blue**.

The authors of the paper propose an approach that allows one to analyze from the first principles the emission from a quantum system (atom, molecule, etc) in the presence of a strong laser field. Specifically, this approach is implemented for analyzing the high order harmonics generation (HHG) from a model atom. The approach is based on the ab initio consideration of the interaction of a quantum-mechanical system (atom, molecule ...) with the modes of a quantized field. The intense laser field acting on the system principally can be treated also as quantum object. But from the very beginning of the paper laser field considered from the standpoint of classical theory. In this case, the interaction of an atom with an electromagnetic vacuum is considered in the first order of the perturbation theory (i.e., only single-photon excitations of field modes appear to exist), while the action of a strong laser field is accurately taken into account. I believe that the development of this approach is fundamentally important, because any radiation of a quantum system at frequencies that do not coincide with the frequency of the acting field always starts with spontaneous emission, i.e. it is the result of interaction with the electromagnetic vacuum surrounding the atom. I am not sure, that is possible to study the appearance of such frequencies in the radiation spectrum, since the semiclassical theory, which is commonly used to analyze the radiation emission from an atom at different frequencies (and in particular, the HHG), is not completely satisfactory (see, for example, Laser Phys. Lett., V. No. 4, 045301 (2016)). I believe that a link to this work in the text of the paper would be more than appropriate.

In this sense, the stated research topic is fundamentally important. Also important is the analysis of the radiation emission and HHG from an ensemble of atoms. Such analysis is especially important for practical applications as in real experiments the emission is observed from the atomic ensemble but not a single atom. The specific features of extremely high order harmonics emission are analyzed beyond the dipole approximation. The possibilities of experimental observation of the effects arising in the proposed approach that are principally cannot be understood within the frames of a semiclassical consideration of the problem, are discussed. Attention is drawn to the fact that in the process of interaction the system "atom + quantized field modes" turns out to be entangled, and, therefore its wave function cannot be factorized into the wave functions of the atomic and field subsystems.

We are very grateful to the referee for these positive comments. We also believe that this is a fundamentally important research avenue that may have a significant impact on the field. Following the referee's suggestion, we have improved our discussion about the ineffectiveness of semi-classical approaches and clarified the differences between the different approaches. We also cite the suggested references and explain their relevance to our manuscript. For example, in the text in the page 2, we now explain:

Such an approach is semi-classical, which is not completely satisfactory and provides a consistent result with full quantum-electrodynamical calculations only in limited cases as was shown in [18].

However, there is a number of fundamental remarks on the paper material.

1) First of all, it should be noted that the approach formulated in the paper regarding the consideration of a single atom dressed by the classical field and interacting with an electromagnetic vacuum (this interaction is taken into account in the first order of the perturbation theory) is actually (in its physical essence) equivalent to the approach proposed in [20,21] from the list of references. I believe that this statement should be clearly stated in the paper under review. Otherwise, if the authors do not agree, it should be clearly stated what is the essential difference between the proposed approach and the approach described in (EPL, V.116, No. 1, P. 14003 (2016); JETP, V. 125, No. 4, P. 587-596 (2017)). The second of these papers should also be cited. In this sense, the novelty of the reviewed paper is the possibility of analyzing of the laser field also in the frames of quantum electrodynamics. This can be important if, for example, the atomic dynamics in "squeezed" electromagnetic fields is analyzed. However, the authors only declared but did not consider in detail such quantum approach to the description of the laser field. I would also mention that the typo in the surname of one of the authors [20,21] exists.

We thank the referee for this comment. We are also grateful for the suggestion of the second reference that we now cite together with the previously cited references by the same group (there, we also cite the Laser Phys. Lett. paper suggested below by the referee). We agree that the approach in these references has several important similarities and important differences and that clarifying them improves our manuscript – we revised the manuscript to explain the comparison.

Specifically, the most important similarity is the treatment of the HHG radiation emission as spontaneous emission by the time-dependent dressed electron states. We added details on this approach in the same place where we originally cited these related papers. We also provide more details in the introduction and in Section II.

There are important differences in the new approach we present. Firstly, as the referee also emphasized, we allow for the driving field to be treated within quantum electrodynamics – and indeed this adds to the novelty of our work. Secondly, in our approach, we do not use the dipole approximation for the emitted HHG field. Thirdly, we expressed the HHG emission in terms of time-dependent transition dipole moments (given by $d_{ji}(t)$ in Eq. (4) in the manuscript). This gives a connection with known formulas for spontaneous emission, and also allows to separate the coherent and incoherent parts of the emission.

We now further clarify all of these points in the manuscript (page 2-3):

More recent studies [21, 22, 23] developed a quantum formalism for HHG in which the electron states are dressed by the driving classical field, and the radiation is seen as spontaneous emission from these time-dependent dressed states.

And page 4:

The formalism constructed here is based on the idea of using quantum electrodynamical perturbation theory for bound electrons that are dressed by an external field (SFQED), similar to the approach described in [21, 23]. We go beyond the previous literature by considering fully

quantized electromagnetic fields (for both the driving and emitted fields) and also taking into account beyond-dipole corrections.

We have also corrected the formatting of refs. [20, 21] (In the revised manuscript [21, 22]).

2) The authors focused on the analysis of the spectrum of spontaneous emission (HHG) arising from the interaction of the Ti-Sa laser radiation with a model one-dimensional helium atom. However, a similar problem was already considered earlier in (Laser Phys. Lett., V.14, No. 5, 055301 (2017); JETP, V. 125, No. 4, P. 587-596 (2017)), but for a model one-dimensional xenon atom. It seems to me absolutely necessary, however, to compare the results of calculations performed in the present paper with those published earlier, at least at a qualitative level.

We thank the referee for bringing these two references to our attention. The revised manuscript now makes a qualitative comparison between the results in the mentioned references and our results (in Fig. 2). We also developed a detailed mathematical comparison between the two approaches, which is now provided in a new supplementary section (SM, Section 10), with the key steps copied below.

Due to the different approaches there are some differences in the numerical results.

- (i) The approach of the Laser Phys. Lett. paper allows to find the HHG radiation for each specific wavevector \mathbf{k} and specific polarization σ – then the integration over the directions of \mathbf{k} should be performed numerically. In our manuscript (detailed comparison below), this integration is performed analytically, however we have to numerically sum/integrate over different contributions to the HHG radiation from transitions elements d_{ji} , where $j \neq i$. Moreover, we integrate the result over all time (from $-\infty$ to $+\infty$), while the Laser Phys. Lett., paper finds corrections for finite times. All these differences in physical parameters and differences in numerical approaches result in different numerical results. Each approach has pros and cons that depend on the desired measurable quantities.
- (ii) We have different parameters in the system. In the Laser Phys. Lett. paper the authors assume an intensity of 10^{14} W/cm², while we assumed an intensity that is 2 times larger. Moreover, we use a 1D helium atom model and they use a 1D xenon atom. These differences resulted in different cut-off energies: in our case, the cut-off is around the 60th harmonic, while in the Laser Phys. Lett. paper the cut-off is around the 23rd harmonic.

These differences also contribute to qualitative corrections: due to our choice of parameters, we have effects beyond the dipole approximation. Moreover, in the Laser Phys. Lett. paper the background radiation (the radiation between integer peaks) is significant, and in our case it becomes dominant (Fig 2 (b) in our manuscript).

We add this discussion to our manuscript, page 11:

Part of the features we presented in Figure 2 can be corroborated by comparing with earlier work. Specifically, quantum corrections arising from transition elements d_{ji} were studied for a single xenon-like atom in [23, 34]. There is an increase of background radiation (emission not at the integer harmonics) especially around low harmonics, which was predicted in [34] as well as in our study. This effect was shown to significantly change the spectrum for strong fields for which the cut-off is approximately at the 20th harmonic [34]. Our work shows that the quantum corrections can completely dominate the spectrum of emission for stronger fields for which the cut-off is approximately at the 60th harmonic. Beyond this comparison, each work emphasizes

additional features, and use a different formalism. We compare the two approaches quantitatively and prove mathematical connections in SM, Section 10.

Supplementary Materials, Section 10:

10. Comparison and proof of equivalence of two approaches for calculating the emission spectrum

In this section, we compare our formalism with the derivation in the paper [12], which calculates radiation emission from a bound electron in a strong field in a different approach than ours. According to Eq. (4) and Eq. (11) in this paper, we have the Schrodinger equation:

$$i\hbar \frac{\partial}{\partial t} |\Psi\rangle = (H_0(t) + V_{\text{int}}) |\Psi\rangle,$$

where H_0 is the Hamiltonian including the classical part of the field, and V_{int} is the perturbation connected with the quantum part of the field. In our notions, the interaction part looks like:

$$V_{\text{int}} = \frac{q}{m} \mathbf{A}_q \cdot \mathbf{P},$$

where in dipole approximation

$$\mathbf{A}_q(r, t) \approx \sum_{k\sigma} \sqrt{\frac{\hbar}{2\varepsilon_0 V c k}} [\mathbf{e}_\sigma a_{k\sigma} e^{-i\omega_k t} + \mathbf{e}_\sigma^* a_{k\sigma}^\dagger(k) e^{+i\omega_k t}] \text{ and } P(t) = p - qA_c(t).$$

According to Eq. (6) in [14], unperturbed state evolves by ordinary TDSE:

$$i\hbar \frac{\partial}{\partial t} |\psi_i\rangle = H_0(t) |\psi_i\rangle.$$

Then, the perturbed state can be found in the following form according to Eq. (9) and Eq. (1) in [12]:

$$|\Psi\rangle = |\psi_i(t)\rangle \otimes |0\rangle + \sum_{k\sigma} |\psi_{k\sigma}\rangle \otimes |k\sigma\rangle.$$

In the 1st order, we can write the following equation for $|\psi_{k\sigma}\rangle$ (Eq. 12):

$$i\hbar \frac{\partial}{\partial t} |\psi_{k\sigma}\rangle = H_0(t) |\psi_{k\sigma}\rangle + \frac{q}{m} \sqrt{\frac{\hbar}{2\varepsilon_0 V c k}} [\mathbf{e}_\sigma^* \cdot \mathbf{P} e^{+i\omega_k t}] |\psi_i\rangle.$$

Hence, [12] suggests firstly to solve TDSE for $|\psi_i\rangle$ (Eq. 6), then to solve TDSE for each \mathbf{k} and σ for the perturbation $|\psi_{k\sigma}\rangle$ (Eq. 12 in [12]) and then find the emission by the formula (Eq. 13 in [12]):

$$\varepsilon = \sum_{k\sigma} \hbar\omega \langle \psi_{k\sigma} | \psi_{k\sigma} \rangle.$$

In comparison, our approach is to solve TDSE $i\hbar \frac{\partial}{\partial t} |\psi(t)\rangle = H_0(t) |\psi(t)\rangle$ not only for the initial state, but also for all orthogonal states, thus we form a full orthogonal basis $\{|\psi_j(t)\rangle\}$. To prove the equivalence of the two approaches, we decompose the state $|\psi_{k\sigma}\rangle$ in time dependent basis:

$$|\psi_{k\sigma}\rangle = \sum_j C_j(t) |\psi_j(t)\rangle,$$

with the equation for the $C_j(t)$ coefficients being the following:

$$i\hbar \frac{\partial}{\partial t} C_j(t) = \frac{q}{m} \sqrt{\frac{\hbar}{2\varepsilon_0 V c k}} \langle \psi_j | \mathbf{P} | \psi_i \rangle \mathbf{e}_\sigma^* e^{i\omega_k t}.$$

Solving this equation we get the following result:

$$|\psi_{k\sigma}\rangle = \frac{q}{m\hbar i} \sqrt{\frac{\hbar}{2\varepsilon_0 V c k}} \sum_j \int d\tau (\mathbf{P}_{ji} \cdot \mathbf{e}_\sigma^*) e^{i\omega_k \tau} |\psi_j\rangle.$$

Then finally, using the equation for the emission (Eq. 13 in [14]) we get:

$$\varepsilon = \frac{q^2}{2m^2 \varepsilon_0} \sum_j \sum_{k\sigma} \frac{1}{V} \left| \int d\tau (\mathbf{P}_{ji} \cdot \mathbf{e}_\sigma^*) e^{i\omega_k \tau} \right|^2,$$

which is exactly Eq. (S8) in our supplementary. Thus, both approaches are mathematically equivalent.

3) In addition to remark 2. Not only the frequency of the incident laser radiation (Ti-Sa laser) and its intensity (2e14 W/cm2), but also the pulse duration and its envelope should be clearly indicated in the paper. Without such data, the data presented in the paper (see Fig. 2) and the corresponding explanations in the text seem to be incomplete.

We thank the referee for this comment. We have now included these details in Figure 2 and 3 in the main text.

4) The authors claim that they describe the dynamics of the atomic system in a classical field analytically. I would like to see how this is done? Is it the SFA approximation? And is it possible to do this for an arbitrary envelope of a laser pulse and its duration, in particular, under conditions of a substantial depletion of the ground (initial) state of an atom?

We agree with the referee that this point needs further clarification. Note that we have purposefully avoided emphasizing a particular numerical technique to obtain the electronic part of the wavefunction. The expression for the radiated spectrum is analytical given d_{ji} , and we did not specify one specific way for how to find d_{ji} . This was done in order to keep the approach (and provided formulas) as general as possible. One may obtain the actual values of transitions matrix elements d_{ji} in a variety of ways according to different levels of theory and approximations, e.g. semi-analytically using the SFA, by directly solving the TDSE, by using TDDFT, and so on. To provide concrete examples in the figures, our numerical calculations directly solved the TDSE by assuming a single active electron in the 1D model of the Helium atom. We intentionally chose the most conventional approach from the HHG community to emphasize that it can be used as is to make quantum predictions as well – this is now clarified in the main text.

We have now added additional details on this to the text (page 9):

We can find $\mathbf{d}_{ji}(t)$ by solving the TDSE for each j , which can be done using any of the previously developed approaches [17,29,30]. We note that several of the qualitative conclusions we draw do not depend on the precise method used to obtain $\mathbf{d}_{ji}(t)$. This way, *all the previous techniques developed for HHG* (e.g., fully numerical approaches, strong-field approximation, etc.) *can be facilitated for studying quantum effects*.

5) Undoubtedly, from a practical point of view, an important element of the paper is the transition to a description of the radiation from an ensemble of atoms. But this transition was made somehow unintelligible with references to rather old publications that are not directly related to the problem. In such a situation a quantum system has become more complex: it is already a number of emitting particles (this is equivalent simply to increasing of the dimension of the space in which the atomic wave function is given). And to describe the formation of a macroscopic radiation field, we must abandon the single-photon approximation for radiation field and consider the possibility of emitting a certain number of photons. Under definite conditions such an approach, in particular, would allow the authors to describe the effect of superradiance of atoms dressed by the field. The effect, which is essentially quantum in nature. It seems to me that the authors here should rework the text of the paper in this direction, and not only speculate enough about proportional terms N^2 and N .

This is a very important point and we were glad to answer it by expanding the discussion of this issue in the revised manuscript. We also agree with the referee that the theory of HHG for an ensemble of atoms is crucial from both an applied and fundamental point of view. To the best of our knowledge, currently, there is not yet a fully quantum theory that connects HHG and superradiance. Regardless of these points, we completely agree with the referee that further explanation for the case of an ensemble of atoms should be provided. Accordingly, we have *substantially* expanded the discussion and justification of this issue in the manuscript. For further explanation of all these effects, we included a new section to the Supplementary (SM, Section 9).

We also made several changes in the manuscript, at the following pages.

In the introduction, page 3:

We describe the transition between the single- and the many-atom regime and under which conditions it happens.

Page 12-13:

We now discuss the emission from many (N) atoms. The emission of an ensemble of atoms can always be separated into a coherent and an incoherent part. Particularly, the incoherent part of the emission is proportional to N , while the coherent part is proportional to N^2 . In standard conditions of HHG experiments, the coherent parts of the emission only arise from the \mathbf{d}_{ji} elements, and all have trivial quantum properties such as Poissonian statistics (SM, Section 5) and a factorizable (non-entangled) atom-field wavefunction. At the same time, the incoherent parts of the emission that arise from the \mathbf{d}_{ji} elements contain features of entanglement and non-Poissonian statistics of light. The effects of incoherent contributions appear in many areas of physics and have been investigated for several decades (e.g. [35]). For a large enough ensemble, the incoherent part of the emission becomes negligible compared with the coherent part. It is a question of fundamental interest to find avenues by which SFQED could make the normally

incoherent parts become stronger and thus reach new quantum effects of many-body systems with nontrivial photon statistics.

We provide a qualitative argument to expose the coherent and the incoherent parts of the radiation. Let us consider N non-interacting atoms in a small volume, which interact with an external driving laser field, neglecting other interactions for the duration of the driving field (see SM, Section 9). In the conventional case, if all the atoms were initially in the ground state, the contributions of the \mathbf{d}_{ij} element of Eq. (4) from different atoms add up coherently, and the sum is proportional to N^2 . In contrast, contributions of all the other elements (involving states distinct from the initial state) from different atoms add up incoherently and the sum is proportional to N . For sufficiently large values of N , we can eventually neglect the incoherent parts, and then, the many-atom HHG is adequately captured by Eq. (5) multiplied by N^2 , in *exact* agreement with the conventional classical theory [2,6,7]. We derive quantitatively this in the SM, Section 9.

Nevertheless, there exist conditions under which the quantum nature of the interaction becomes significant even for a large ensemble. Notably, in our numerical example in Figure 2 the magnitude of $|\mathbf{d}_{21}|^2$ (the largest matrix element) is 10^4 times greater than the magnitude of $|\mathbf{d}_{11}|^2$ (Figure 2). Hence, when the number of active atoms is $N < 10^4$, we expect significant observable deviations from the conventional HHG theory Eq. (5), i.e., “quantum corrections”. This scenario contains features that cannot be explained by the semi-classical approach, and it thus underscores the importance of the SFQED approach and shows perspectives for the creation of macroscopic (at least mesoscopic) quantum states of entangled light and matter. A full investigation of quantum many-body effects of SFQED is left for future work.

Page 20:

From a fundamental standpoint, finding truly quantum effects in *ensembles* of atoms will open new perspectives for HHG and many other non-linear optical effects. While a complete formulation of many-body HHG is still elusive, it may reveal promising opportunities to enable truly many-body quantum electrodynamical effects with a large number of photons, which would be surprising from a fundamental point of view.

Regarding reference [35] (in previous version of the manuscript [32]), we cited it because it explains why there is an incoherent part to the emission in the first place (in a general scenario, not only when driven by a strong field). Given the new discussion and derivation in the revised manuscript, it is now of a somewhat lesser importance. If the referee recommends we do so, we will remove it altogether.

To conclude, in my opinion the material presented is important and interesting, but is needed in significant revision.

We are grateful for the referee for his supportive comments. We believe that the revised manuscript is now substantially improved, and thank the referee for helping us make these improvements. We hope that the referee will now fully support the publication of our work in light of these revisions.

Response to referee #2 comments

I recommend publication of “On the Quantum-Optical Nature of High Harmonic Generation,” by Gorlach et al. This well-written manuscript highlights a program of analyzing laser high harmonic generation from the standpoint of QED. The authors couple a single-active-electron one-dimensional version of helium to the quantized electromagnetic field following a perturbative approach. They comment on a host of quantum effects that can be seen using their model. Each of these effects might be the subject of a separate publication. By commenting on all of them, the authors apparently feel that this work merits publication in Nature Photonics. I tend to agree. In this case, their overall approach to the problem is emphasized, illustrating the variety of effects that can be studied with it.

I found the discussion surrounding Fig. 2 particularly interesting. The authors make it clear that, what researchers using the semiclassical approach often loosely call “single-atom” calculations, is really an ensemble average of atomic dipoles in a region of space small compared to a laser wavelength. They show how the true single-atom spectrum shows a continuum of wavelengths, whereas ensemble phase matching pulls out the discrete dipole spectrum.

The authors also find that individual photons should carry information about the entire high harmonic spectrum. They propose a basic photon correlation experiment to demonstrate this. To realize this experiment, it may be difficult to find an attenuator that transmits a wide spectrum uniformly. Still, the concept is interesting.

The authors comment on the failure of the dipole approximation for the very high-order harmonics, when the emission wavelength is no longer much larger than the trajectory path of the electron. In that case, the quadrupole term becomes important.

I have analyzed this manuscript as a person with a background in high harmonics, not as an expert in QED. I am unable to judge technical soundness of the QED calculations.

We are very grateful the referee for taking the time to review our manuscript. We wish to thank the referee for the encouraging and positive response, and for recommending the publication of our paper in its current form. Thank you.

We want also to mention that following this round of review, the manuscript was also improved in several technical aspects. We included more discussions about the many-body implications of HHG, as well as provided further formal justification to the separation of the emission from an ensemble of atoms to coherent and incoherent parts. We believe that these changes have improved the manuscript.

Response to referee #3 comments

We thank referee for taking time to review our manuscript. However, we wish to point out that we strongly disagree with the referee's comment about the novelty of our work. The referee's comment about novelty differ also from the opinions of the other referees.

Before addressing all of the referee's comments, we wish first to briefly address this main point of criticism and point out the main novel results presented in our work. Firstly, we built the general theory of quantum electrodynamical processes in strong fields (SFQED), considering emission induced by the strong field as spontaneous emission process from field-dressed atomic states. Secondly, we analyzed in details the new consequences to HHG that were revealed by this general approach, showing that quantum corrections from a single atom could completely change the emission spectrum. Thirdly, we demonstrated that these quantum corrections lead to non-trivial photon statistics in HHG – entanglement and non-Poissonian statistics. We predicted specific regimes where these effects can lead to detectable variations in the emission spectrum. Lastly, we formulated for the first time how beyond-dipole effects due to the *emitted field* manifest as corrections to the HHG emission, which in some regimes are larger than the conventional corrections arising due to beyond dipole terms due to the *driving field*. To summarize, we believe that our work has fundamental importance that was not previously reported in the literature. We compare each of the above points with the two references suggested by the referee and with the rest of the literature in the HHG community – emphasizing the novelty of our work.

Having stated this, we now respond to the referee's comments point-by-point, where his/her comments are denoted by **Bold**, our answers denoted by ordinary text and changes in the manuscript are written in blue color.

The title and half of the abstract of the present manuscript are very exciting. One expects to see in the paper, as stated in the first half of the abstract, investigation of the statistical properties of the emitted HHG light and learn how and at which conditions a squeezed HHG light can be created. Unfortunately, the text of the paper brings disappointment. The paper put forward three main results:

- 1) Each HHG photon is a spectral comb**
- 2) Incoherent HHG can be emitted when recombination occurs to the bound state different than the original state**
- 3) For HHG at very high energies, nondipole emission should be accounted for.**

We respectfully disagree with the referee about what are the main contributions of our manuscript.

We wish to point out that referee #3 did not refer to the importance of the strong-field QED (SFQED) theory that we developed, which goes beyond high harmonic generation. It is the first time that a fully quantized strong-field quantum theory is developed (where both the driving field and the emitted field are quantized) and provides concrete predictables. We would like to point out that this point about the fully-quantum theory is exactly what was emphasized as the important part of the work by referees #1 and #2.

Our theory is applicable in a wide range of phenomena in nonlinear optics, such as bound and free electrons irradiated by strong fields, non-linear Compton scattering, electrons propagating in time varying medium, etc. Consequently, it could find wide use in the broad community interested in quantum nonlinear optical effects, and not only in HHG.

Another important result in our work, which is at least as important as points 1-3, is the finding of entanglement between the photon emission and the emitting atoms.

The revised manuscript now better emphasizes the main contributions beyond points 1-3 by changes in the introduction and in the rest of the text. We thank the referee for raising these points that helped clarify important aspects of our work.

The HHG community is well aware of all three points.

We respectfully disagree with the referee about the assessment of the novelty and importance of the three points above. Below, we discuss them in more details.

Moreover, the second and third points are not directly related to the quantization of the emitted photon field.

Regarding the second point, the correct treatment of incoherent radiation cannot be disconnected from the field quantization, which is a new consideration in our manuscript. The findings we presented in Sections II and III and in Fig. 2 exactly show quantitative predictions that arise from the field quantization. To the best of our knowledge, there is no way to get this result without the field quantization.

Moreover, we believe that there is a value also in highlighting the part of the results that are found to be equivalent to the conventional derivation (e.g., Eq. (5) in the manuscript).

Regarding the third point, we agree that it could have been developed without the quantization of the emitted field. *However*, note that this is concluded in hindsight after the beyond-dipole result was developed and presented in our work for the first time.

In addition, we would like to strongly emphasize that the quantization of the emitted photon field is a key point in our manuscript, and is intimately related to results beyond points 1-3. Specifically, it leads to features of entanglement and non-Poissonian statistics for the incoherent part of the HHG emission associated with transitions to dressed states different from those associated with the ground state.

To emphasize the importance of the quantization of the field we add the sentence in page 4 in the manuscript:

The quantization of the electromagnetic fields enables analyzing the quantum statistics of the emitted field and of the driving field.

Following the referee's comment on this, the improved manuscript gives additional details about the entanglement, as well as the quantum nature of the incoherent part of the radiation.

On the first point, it is well accepted that the HHG emission is a coherent process and the emitted radiation field is a coherent field. The coherent harmonic field described in the quantum language is, in fact, equivalent to the collection of photons each of which is a spectral comb.

The referee is correct that this is the current consensus in the HHG community – that HHG emission is coherent. However, our work shows that this is not always the case, and that the regimes in which the coherent nature of HHG is broken involve new quantum effects. The quantum theory that we developed with the SFQED formalism is necessary for making such predictions.

We explicitly show that the field can only be coherent in the many-atom regime, but it also always carries an additional incoherent part (due to quantum corrections), which affects the quantum state of the emission and the photon statistics. Moreover, note that there is no proof that in the many-atom regime, HHG will always be coherent, and we point out specific avenues by which even many atoms could emit light with different statistics and significant entanglement.

We now further emphasize this in page 12 in the manuscript:

We now discuss the emission from many (N) atoms. The emission of an ensemble of atoms can always be separated into a coherent and an incoherent part. Particularly, the incoherent part of the emission is proportional to N , while the coherent part is proportional to N^2 . In standard conditions of HHG experiments, the coherent parts of the emission only arise from the \mathbf{d}_{ji} elements, and all have trivial quantum properties such as Poissonian statistics (SM, Section 5) and a factorizable (non-entangled) atom-field wavefunction. At the same time, the incoherent parts of the emission that arise from the \mathbf{d}_{ji} elements contain features of entanglement and non-Poissonian statistics of light. The effects of incoherent contributions appear in many areas of physics and have been investigated for several decades (e.g. [35]). For a large enough ensemble the incoherent part of the emission becomes negligible compared with the coherent part. It is a question of fundamental interest to find avenues by which SFQED could make the normally incoherent parts become stronger and thus reach new quantum effects of many-body systems with nontrivial photon statistics.

The coherent harmonic field described in the quantum language is, in fact, equivalent to the collection of photons each of which is a spectral comb.

To the best of our knowledge, the description of the single photon of HHG was not investigated before in any previous work. If the referee knows such works, we would appreciate if they refer us to the relevant literature.

More importantly, our work actually shows that the quantum nature of each single photon in HHG is more intricate than just being a spectral comb. *Only under certain conditions*, the collection of photons can be described by a multiplication of coherent states of different

frequencies, and only then, each single photon can be thought of as a spectral comb. Such conditions can be achieved, for example, if the system starts at the ground state in the many-atom regime and under additional approximations (such as the dipole approximation).

More generally, the SFQED formalism we developed allows us to identify the resulting general photonic state (Eq. (3) in the manuscript), which *is generally not coherent*, thus each photon does not only contain a spectral comb of coherent HHG field, but also includes incoherent contributions, which lead to such effects as entanglement. We suggested an experimental setup that can test this concept.

To summarize, this more general type of photonic state is the accurate way of describing the nature of each single photon in HHG. Most interestingly, the quantum treatment shows that HHG can produce different photonic states, including ones different from a coherent state.

Following the referee's comment on this, we emphasize this point in the revised manuscript in page 8:

Thus, from Eq. (3) we conclude that the wavefunction $|\Psi(t)\rangle$ not only shows that each photonic state is a superposition of multiple frequencies, but also shows that it has quantum features such as entanglement and carries the information about all transition matrix elements d_{ji} .

Our manuscript now better distinguishes the differences in the photon statistics, and especially the cases where the photons are a multiplication of coherent states (see changes in pages 12-13).

The second point not only well known but it is generally not used in applications, because it creates incoherent radiation which is minor if the medium is not very dilute.

- We completely disagree with referee that our results are well known. We point out the following considerations that relate to the second point and show what is novel about it: the quantum corrections we found give *different quantitative* results comparing with the conventional approach, and these corrections were not studied before.
- These quantum corrections are not necessarily minor. They *can be observed*, for example by (1) employing a pulse on-resonance with an atomic transition before applying the strong driving field or (2) exploring HHG with a small amount of atoms (a few thousands), or a small solid-state target (e.g., a nano-flake). Thus, the quantitative description of the transitions to different bound states is important.
- These corrections have *direct applications* in quantum optics, because of their non-trivial statistics. Moreover, such corrections are observable and may be connected with the latest experiments in the field of Hyper Raman Lines [e.g., Etienne Bloch et al. *New J. Phys* **21**, 073006 (2019)].

Our work about the quantum correction made new predictions that are different from all the work done before. If the referee is familiar with other papers with similar results, we would be glad to compare our results with them.

To better understand the novelty in our work, it is beneficial to compare our approach with the conventional description of incoherent radiation. Usually, incoherent radiation is understood as spontaneous recombination after the recollision of the electronic wave packet, which can be neglected due to the random phase of the spontaneous recombination amplitude [Kohler et al,

Advances In Atomic High-Harmonic Generation, 61, 159-208 (2012)]. While this description works in many scenarios, the full SFQED theory allows us to make additional predictions that cannot be captured by assuming random phases. For example, the SFQED approach enables to predict what will be the HHG radiation from a single atom or from a few atoms (or even up to 10^4 atoms as our numerical example shows).

In the revised version of the manuscript, we also added the qualitative description of incoherent and coherent emission in the framework of full quantum electrodynamics (SM Section 9).

Moreover, following the comments by the referee, we modified the manuscript to clarify the differences between our approach and the conventional treatment of the incoherent parts of HHG. We now specifically emphasize in what ways our predictions are novel.

Page 12-13:

We provide a qualitative argument to expose the coherent and the incoherent parts of the radiation. Let us consider N non-interacting atoms in a small volume, which interact with an external driving laser field, neglecting other interactions for the duration of the driving field (see SM, Section 9). In the conventional case, if all the atoms were initially in the ground state, the contributions of the \mathbf{d}_{ij} element of Eq. (4) from different atoms add up coherently, and the sum is proportional to N^2 . In contrast, contributions of all the other elements (involving states distinct from the initial state) from different atoms add up incoherently and the sum is proportional to N . For sufficiently large values of N , we can eventually neglect the incoherent parts, and then, the many-atom HHG is adequately captured by Eq. (5) multiplied by N^2 , in exact agreement with the conventional classical theory [2,6,7,36]. We derive quantitatively this in the SM, Section 9.

Nevertheless, there exist conditions under which the quantum nature of the interaction becomes significant even for a large ensemble. Notably, in our numerical example in Figure 2 the magnitude of $|\mathbf{d}_{21}|^2$ (the largest matrix element) is 10^4 times greater than the magnitude of $|\mathbf{d}_{11}|^2$ (Figure 2). Hence, when the number of active atoms is $N < 10^4$, we expect significant observable deviations from the conventional HHG theory Eq. (5), i.e., “quantum corrections”. This scenario contains features that cannot be explained by the semi-classical approach, and it thus underscores the importance of the SFQED approach and shows perspectives for the creation of macroscopic (at least mesoscopic) quantum states of entangled light and matter. A full investigation of quantum many-body effects of SFQED is left for future work.

Page 20:

From a fundamental standpoint, finding truly quantum effects in *ensembles* of atoms will open new perspectives for HHG and many other non-linear optical effects. While a complete formulation of many-body HHG is still elusive, it may reveal promising opportunities to enable truly many-body quantum electrodynamical effects with a large number of photons, which would be surprising from a fundamental point of view.

The authors' contribution is that they write a formal wave function in Eq.(3) and indicate on the entanglement between the atom and the photon field. However, the reader expects from the paper to learn about “intriguing consequences in the field of quantum optics” which emerge due to the entanglement. It is stated in the paper but not shown.

We thank the referee for raising this comment. We expanded the discussion about the entanglement and its possible consequences in the revised manuscript. (We also changed the wording to avoid the used of the subjective word “intriguing” in relation to our findings).

Beyond clarifications in places where the entanglement was previously mentioned, we added a further discussion at pages 8 of the manuscript. We copy part of it here.

The entanglement implies that there remains a connection between the photon and the emitting atom after emission even in very strong fields. This result is significant from a fundamental point of view, because entanglement between an atom and a field, or between multiple photons, is usually only found in a weak perturbative regime (e.g., spontaneous parametric down conversion by the perturbative $\chi^{(2)}$). However, here we showed that entanglement can exist even in strong non-perturbative fields, which further motivates the study of HHG with the full-quantum description.

Page 20:

Looking forward, one of the most important new aspects our formalism reveals is the *entanglement* between the photons and the emitting atoms, which remains after the emission even in very strong fields. This entanglement may have important consequences in the field of quantum optics and in pump-probe experiments.

Finally, on the third point, the nondipole effects at a photon emission will be important at photon energies about 10 keV, to have the wavelength of the emitted photon comparable to the atomic size. This is already becoming achievable in experiments and nondipole effects are discussed in many publications, see e.g. M. Klaiber, et al., PRA 75, 063413 (2007), and a review [M. C. Kohler, et al]. Frontiers of atomic high-harmonic generation], which includes many other citations on this topic.

We thank the referee for this comment, however there seems to be a misunderstanding. Our manuscript distinguishes *two different types* of non-dipole corrections: (1) *due to the emitted field* and (2) *due to the driving field*. Both effects are completely unrelated, and are only connected by name.

The references provided by the referee [M. Klaiber, et al., PRA 75, 063413 (2007); Kohler et al, Advances In Atomic High-Harmonic Generation, 61, 159-208 (2012)] – now [36] and [44] in the revised manuscript as well as additional references that we cited [17, 37, 38, 39] – all of them only refer to beyond-dipole corrections stemming from the *driving field*. But in the manuscript, we discuss the corrections due to the *emitted field*, which were not discussed before to the best of our knowledge. Thus, the beyond-dipole effects due to the *emitted field* are not well known. Their implications for the theory of HHG are all new.

A detailed discussion of differences between the dipole approximation for the emitted field and for the driving field is provided below. We also modified the manuscript to better explain the differences between the two types of dipole approximations. We now emphasize the critical role of each effect to prevent the potential confusion between the two.

Let us emphasize the differences by elaborating on the beyond-dipole corrections due to the *driving field*, which were studied before in the literature. These corrections are fundamentally different because they can change the way we find the electron wavefunction $|\phi(t)\rangle$, but they do not change the form of Eq. (5) that governs the conventional emission. i.e., they do not modify the expression $\langle\phi(t)|\mathbf{d}|\phi(t)\rangle$. In contrast, the beyond-dipole correction due to the *emitted field* modify this expression and thus change the form of Eq. (5) entirely.

The corrections due to the *driving field* include the dependence of the potential $\mathbf{A}(\mathbf{r}, t)$ on the distance \mathbf{r} as well as all relativistic corrections to the $|\phi(t)\rangle$, which are discussed in [M. Klaiber,

et al., PRA 75, 063413 (2007) – now [44]. In our revised manuscript, we emphasize that such beyond-dipole corrections are known and cite the relevant literature.

The reason for the beyond-dipole corrections due to the emitted field is that the wavelength of the *emitted* light itself (not the driving field) is comparable to the size of the atom, and for short wavelengths, the emission must be described by Eq. (6) instead of Eq. (4). The corrections connected with differences between Eq. (6) and Eq. (4) are what we call “beyond-dipole corrections due to *the emitted field*”.

We show that these beyond-dipole corrections can be much larger than the corrections discussed before (due to the *driving field*).

We have incorporated these remarks, in page 14 of the manuscript:

We want to clarify the difference between the two different types of beyond-dipole corrections, which are fundamentally different and are related only by name. Beyond-dipole corrections due to the *driving field* change the TDSE and the resulting wavefunction $|\phi(t)\rangle$, but do not change the form of Eq. (5). Such corrections to the evolution of the wavefunction include relativistic corrections to the Schrodinger equation (e.g., [36, 44]). In contrast, the beyond-dipole corrections to the *emitted field* described in this work do not change the wavefunction in the matrix element, but modify the form of Eq. (5), as we see below.

Concluding, the present manuscript contains results of minor novelty and I cannot recommend it for Nature Communications. After significant revision, combining the supplement with the paper, providing detailed experimental setup for measuring the single HHG photon quantum properties, with an appropriate feasibility study, providing an example of application of the atom-photon entanglement, and elaborating more on the nondipole features in the HHG spectra, the paper may be suitable for publication in Scientific Reports.

We hope that our extensive revisions and clarifications will help change the referee’s assessment, as we believe that the negative review results from a misunderstanding of what are the main findings in our manuscripts.

We wish to summarize the main points of novelty that we believe justify its publication in *Nature Communications*:

1. We developed the first fully quantum formalism of HHG, which can be applied for a wide range of additional nonlinear phenomena such as the nonlinear Compton effects, and multiple Thompson scattering, etc. This approach can be used in a variety of physical systems.
2. We show that there are additional quantum corrections to the semi-classical spectrum. We demonstrate that these corrections *completely change* the spectrum of HHG from a single atom. Under certain conditions, these quantum corrections can become significant even for thousands of atoms. Such a prediction has direct applications in quantum optics and related fields, because these quantum corrections are responsible for non-trivial statistics and entanglement.
3. We showed explicitly using a quantum formalism that each photon can become a spectral comb. More importantly, we showed that in general the situation is more complicated and depends on the number of atoms. This point is also applicable to many other nonlinear processes. We further suggested how this prediction can be tested experimentally.

4. We formulated beyond-dipole corrections to the HHG emission *due to the emitted field*. We show that these corrections can lead to large measurable effects in HHG spectra, which were not predicted before.

Reviewers' Comments:

Reviewer #1:

Remarks to the Author:

The authors responded to all my comments. I suppose that the paper can be published in the present form.

Reviewer #3:

Remarks to the Author:

Unfortunately, the authors revised only the verbal part of the paper and not the physical content. The problem with the present manuscript is that its verbal declarations do not fit to the concrete obtained results. In the following, I will respond to the authors' statements.

The authors state: "We wish to point out that referee #3 did not refer to the importance of the strong-field QED (SFQED) theory that we developed, which goes beyond a high harmonic generation. It is the first time that a fully quantized strong-field quantum theory is developed (where both the driving field and the emitted field are quantized) and provides concrete predictables. We would like to point out that this point about the fully-quantum theory is exactly what was emphasized as an important part of the work by referees #1 and #2. Our theory is applicable in a wide range of phenomena in nonlinear optics, such as bound and free electrons irradiated by strong fields, non-linear Compton scattering, electrons propagating in time varying medium, etc. Consequently, it could find wide use in the broad community interested in quantum nonlinear optical effects, and not only in HHG."

When speaking about fully quantized strong-field quantum theory applicable in a wide range of phenomena in nonlinear optics, then one should name pioneering works: J Bergou and S Varro J. Phys. A 14, 1469 (1981); 14, 2281 (1981) [and the related one I. Berson, Zh. Eksp. Teor. Fiz. 56, 1627 (1969)] applied for nonlinear Compton scattering in a quantized laser field.

It is also not true that the authors consider the case of the quantized strong laser field. In fact, the concrete calculation has been done with the electron wave function in the classical laser field $A_c(t)$. This is an example of inconsistencies of statements and real calculations or obtained physical results.

Then, it is not surprising the comment of Referee #1:

"First of all, it should be noted that the approach formulated in the paper regarding the consideration of a single atom dressed by the classical field and interacting with an electromagnetic vacuum (this interaction is taken into account in the first order of the perturbation theory) is actually (in its physical essence) equivalent to the approach proposed in [20,21] from the list of references"

By the way, they do not cite a similar recent fully quantized strong-field quantum theory: N. Tsatrafyllis, et al. Phys. Rev. Lett. 122, 193602 (2019).

The authors state: "Another important result in our work, which is at least as important as points 1-3, is the finding of entanglement between the photon emission and the emitting atoms."

The entanglement between the photon emission and the emitting atoms is declared and given by the formal wave function Eq.(3), but never investigated, and any physical consequences are not deduced.

The authors state: "Regarding the second point, the correct treatment of incoherent radiation cannot

be disconnected from the field quantization, which is a new consideration in our manuscript”

I will cite several papers which are devoted to the problem of HHG with a single atom and to the treatment of coherent and incoherent radiation in the HHG process:

I. R. Senitzky, Phys. Rev. 111, 3 (1958); B. Sundaram and P. W. Milonni, Phys. Rev. A 41, 6571 (1990); *ibid.* 45, 4706 (1992); *ibid.* 47, 1327 (1993).

The authors state: “In addition, we would like to strongly emphasize that the quantization of the emitted photon field is a key point in our manuscript, and is intimately related to results beyond points 1-3. Specifically, it leads to features of entanglement and non-Poissonian statistics for the incoherent part of the HHG emission associated with transitions to dressed states different from those associated with the ground state.”

The emitted photon statistics, in particular non-Poissonian statistics, is not calculated and is not investigated in the paper, it is just declared.

The authors state: “Regarding the third point, we agree that it could have been developed without the quantization of the emitted field. However, note that this is concluded in hindsight after the beyond-dipole result was developed and presented in our work for the first time”

and “We thank the referee for this comment, however there seems to be a misunderstanding. Our manuscript distinguishes two different types of non-dipole corrections: (1) due to the emitted field and (2) due to the driving field. Both effects are completely unrelated, and are only connected by name.”

The nondipole features of HHG radiation is a different topic not related to the field quantization and its brief discussion within a conceptual paper is not relevant. Moreover, the authors were not careful, in the cited papers a fully relativistic theory of HHG were developed, which takes into account as nondipole effects due to strong driving field as well as those due to a high energy photon emission, see e.g. Eq.(5) in Milosevic et al. PRA 63, 011403(R), or Eq.(37) in Klaiber et al. PRA 75, 063413 (2007).

Concluding, the quantized strong-field quantum theories are not new and discussed since the work of Bergou and Varro, recently by the group of A.M. Popov, and very recently, see Refs. [22-26] cited in the paper, and the paper of N. Tsatrafyllis, et al. which I mentioned above. Further, the coherent and incoherent emissions in HHG are also discussed within a quantized field theory in the papers of B. Sundaram and P. W. Milonni, mentioned above. The new discussion in the present paper would be justified only if they were able to calculate some physical properties of entanglement and photon statistics, rather than do verbal obvious statements that the entanglement and photon statistics can be calculated only within quantum theory.

I cannot recommend the paper for publication in Nature Communications.

Response to Reviewer #3

Before the detailed point-by-point response, we wish to first emphasize here that we added new results to the manuscript as requested by Reviewer #3. Specifically, we added results about the photon statistics of HHG. These are the first predictions for such an effect in HHG, showing squeezing and the Mandel parameter (demonstrating non-Poissonian statistics). These help to emphasize the strength of our formalism and the new physics that can be discovered by applying a quantum electrodynamical analysis to study HHG. We believe that these additions answer to the request by Reviewer #3. These new results are presented in two new Figures (3 and 4) inside a new section in the manuscript (IV). We also wrote a new supplementary section (SM, Section 6).

Below, we provide a detailed point-by-point response to all the points raised by Reviewer #3.

Point-by-point response to the comments by Reviewer #3

“Unfortunately, the authors revised only the verbal part of the paper and not the physical content. The problem with the present manuscript is that its verbal declarations do not fit to the concrete obtained results.”

We now add the numerical calculation of the quantum statistics of HHG.

We also note that the manuscript from the previous round included several new sections that contributed to the formalism and added new results. For example, we provided a new derivation regarding the comparison of single-body and many-body HHG, and the comparison of the coherent emission and the incoherent emission. We also added a detailed derivation comparing parts of our computational method with the analogous part in the papers by the Popov group.

“When speaking about fully quantized strong-field quantum theory applicable in a wide range of phenomena in nonlinear optics, then one should name pioneering works: J Bergou and S Varro J. Phys. A 14, 1469 (1981); 14, 2281 (1981) [and the related one I. Berson, Zh. Eksp. Teor. Fiz. 56, 1627 (1969)] applied for nonlinear Compton scattering in a quantized laser field.”

We agree with the reviewer that the provided references describe a quantum model and we included them in the revised manuscript. However, please note that all these references are not directly connected with the formalism derived in our manuscript. Particularly, these three references are about the fully quantum description of a *free electron* interacting with a *monochromatic field*. In contrast, our approach (SFQED) deals with a *bound electron* under an *arbitrary (not only monochromatic)* field. In contrast with the pioneering works above, the SFQED approach suggested in our manuscript can capture the physics of HHG. Since our work considers bound electrons, we can apply it to HHG in many systems including a single atom and many atoms in gas or solid. This important difference required multiple new developments in the formalism that make our contribution to SFQED an important part of our paper.

“It is also not true that the authors consider the case of the quantized strong laser field. In fact, the concrete calculation has been done with the electron wave function in the classical laser field $A_c(t)$. This is an example of inconsistencies of statements and real calculations or obtained physical results.”

We respectfully disagree. Our manuscript describes a fully quantized field and not only a classical laser field. There is no classical field approximation used in our work.

Section 1 of the Supplementary Material presents the derivation of how using a change of variables with a displacement operator splits the fully quantum field into a classical part $A_c(t)$ and a quantum part \hat{A}_q . The key equations arising from this derivation are presented in the main text and constitute a key point of novelty in our work. This section proves analytically that using the displacement operator reduces the problem with the quantum field into a time-dependent Hamiltonian with a classical field.

“By the way, they do not cite a similar recent fully quantized strong-field quantum theory: N. Tzafrafilis, et al. Phys. Rev. Lett. 122, 193602 (2019).”

We respectfully note that this reference [Phys. Rev. Lett. 122, 193602 (2019)] was actually cited in our manuscript (Ref. [26] in the previous version of the manuscript, and now [29]). We also cited other works by the same group in Nat. Comm. [24] and Sci. Rep. [25] (in the current version of the manuscript [27] and [28]).

Let us explain the fundamental differences in the approach in these papers comparing with our SFQED approach. The main idea of [24-26] (in the current version [27-29]) is to use the SFA approximation and the analogue of three-step model (when neglecting the action of the atomic potential). Thus, during the motion of an electron, this theory considers interaction between a fully quantized field and a free electron. Such an approach contains certain quantum features of the field, but due to the used approximations this approach cannot be used to predict the effects of our manuscript (e.g., it is applicable only for very high intensities). Our SFQED approach does not use any approximations of the TDSE but relies on the fully quantum treatment of the electron, which substantially extends beyond the approach in [24-26] (in the current version [27-29]).

“Then, it is not surprising the comment of Referee #1:

First of all, it should be noted that the approach formulated in the paper regarding the consideration of a single atom dressed by the classical field and interacting with an electromagnetic vacuum (this interaction is taken into account in the first order of the perturbation theory) is actually (in its physical essence) equivalent to the approach proposed in [20,21] from the list of references.”

Please note that Reviewer #1 not only compared our work to [20,21] ([24],[25] in the current version) but also emphasized the novelty and importance of our approach in relation to it: *“In this sense, the stated research topic is fundamentally important. Also important is the analysis of the radiation emission and HHG from an ensemble of atoms. Such analysis is especially important for practical applications as in real experiments the emission is observed from the atomic ensemble but not a single atom.”*

Reviewer #1 also supported the publication of our manuscript in *Nature Communications*.

The differences and similarities between our approach and the approach of the Popov's group (Refs [20, 21] in the previous version of the manuscript) were thoroughly discussed in the Supplementary Materials Section 10.

We emphasized the important differences between these two approaches in our previous response:

“There are important differences in the new approach we present. Firstly, as the referee also emphasized, we allow the driving field to be treated within quantum electrodynamics – and indeed this adds to the novelty of our work. Secondly, in our approach, we do not use the dipole approximation for the emitted HHG field. Thirdly, we expressed the HHG emission in terms of time-dependent transition dipole moments (given by $d_{ji}(t)$ in Eq. (4) in the manuscript). This gives a connection with known formulas for spontaneous emission, and also allows to separate the coherent and incoherent parts of the emission.”

Reviewer #1 agreed with our analysis in the previous round.

“I will cite several papers which are devoted to the problem of HHG with a single atom and to the treatment of coherent and incoherent radiation in the HHG process:

I. R. Senitzky, Phys. Rev. 111, 3 (1958); B. Sundaram and P. W. Milonni, Phys. Rev. A 41, 6571 (1990); ibid. 45, 4706 (1992); ibid. 47, 1327 (1993).”

We respectfully note that we already cited and discussed one of these papers in the previous version of our manuscript (discussion at page 12 and Ref [35] in the previous version, which is now [38-41]). The other papers contain the same formalism as the one we cited.

Comparing our work to these references actually helps to emphasize the novel prediction we provide. These papers neglect the incoherent part of the HHG emission, while we show the opposite – we find conditions for which the incoherent part is significant. Specifically:

- The incoherent part can be larger than the coherent part for both a single atom and even for a few thousand atoms. We provide a quantitative example in the manuscript (Figure 2). Thus, our prediction shows the possibility to measure experimentally the incoherent part of HHG.
- The incoherent part of the HHG emission is responsible for atom-photon entanglement and sub-Poissonian statistics (as shown in Figure 3 in the new version of the manuscript). We now also added Section 6 of the Supplementary and Section IV in the main text with Figures 3 and 4, which show non-Poissonian statistics and squeezing in HHG for the first time (exploring both regimes, the coherent and incoherent).

Therefore, our predictions are novel and do not overlap with any of the suggested references.

We now cite all these papers together with reference [35] (appearing as [38-41]).

“The nondipole features of HHG radiation is a different topic not related to the field quantization and its brief discussion within a conceptual paper is not relevant. Moreover, the authors were not careful, in the cited papers a fully relativistic theory of HHG were developed, which takes into account as nondipole effects due to strong driving field as well as those due to a high energy photon emission, see e.g. Eq.(5) in Milosevic et al. PRA 63, 011403(R), or Eq.(37) in Klaiiber et al. PRA 75, 063413 (2007).”

We carefully looked through the two suggested references [Milosevic et al. PRA 63, 011403(R) (2000); Klaiber et al. PRA 75, 063413 (2007)]. Indeed, Eq. (5) in the first reference and Eq. (37) have all non-dipole contributions. However, these equations are *not* what the papers solve. These equations are just setting up the initial problem *before* the approximations applied in the very beginning of the papers. The approximations applied in these papers neglect the terms that are responsible for the effect we predicted in our manuscript.

For example, both references explicitly neglect the even harmonic emission that we predict in Figure 5 (“... *the emission of even harmonics. For the fields we consider we check that its contribution is still negligible*” [PRA 63, 011403(R) (2000)]). The predictions in these papers are the opposite of our finding of significant even harmonics that directly arise from the beyond-dipole contribution to the emission.

The approximations at the core of these two papers [PRA 63, 011403(R) (2000); PRA 75, 063413 (2007)] make it impossible to capture the beyond-dipole effect that we predict, as they only consider on-axis emission for which the effect we predicted is zero. i.e., these papers cannot capture the beyond-dipole emission predicted in our manuscript.

For these reasons, the papers cannot make our prediction for the beyond-dipole corrections. Such claims are not made in these papers. To the best of our knowledge, our work is the first to predict such an effect for HHG.

We list more specific novel results arising from our treatment of beyond-dipole corrections:

- We found the beyond-dipole corrections using the framework of time-dependent Schrodinger equations without approximations and without using semi-classical models (the semi-classical model was used in [PRA 63, 011403(R) (2000); PRA 75, 063413 (2007)]).
- We present an analytical expression for these beyond-dipole corrections and for the angular distribution (not done in these two references).
- We showed the conditions for which these corrections can be observed, e.g. using a thin layer of atoms (nothing of that kind was done before).
- Our numerical results are completely different from what is mentioned in the two references. *Milosevic et al.* only consider intensities of order 10^{16} - 10^{17} W/cm², whereas our work shows that the beyond-dipole corrections can be observed at 10^{14} W/cm². Additionally, even in these intense conditions, the results in these references predict negligible even harmonic generation, while we predict significant emission of even harmonics.

Hence, part of the novelty of our work is in findings the conditions and predicting experimental opportunities for beyond-dipole corrections with ordinary HHG intensities (e.g., 10^{14} W/cm²).

“The entanglement between the photon emission and the emitting atoms is declared and given by the formal wave function Eq.(3), but never investigated, and any physical consequences are not deduced.”

“The emitted photon statistics, in particular non-Poissonian statistics, is not calculated and is not investigated in the paper, it is just declared.”

“The new discussion in the present paper would be justified only if they were able to calculate some physical properties of entanglement and photon statistics, rather than do verbal obvious statements that the entanglement and photon statistics can be calculated only within quantum theory”

Our manuscript now provides the results requested by the referee in this comment.

We now calculated the physical properties of the photon statistics in a new numerical simulation. Specifically, we found the Mandel parameter of the emitted light in the single-atom regime (Figure 3 in the main text) and the Mandel parameter and squeezing of the emitted light in many-atom regime (Figure 4 in the main text). We added a new section (Section IV) in the manuscript, where we discuss the obtained results. All the details about the theory and the numerical calculations can be found in the new section of the Supplementary Material (Section 6).

We believe that these additions answer to the requests by the reviewer.

Reviewers' Comments:

Reviewer #1:

Remarks to the Author:

As before, I recommend the paper for publication. But now I strongly recommend to cite and briefly discuss one more very recent paper concerning the problem under the study (Yangaliev D N, Krainov V P, and Tolstikhin O I 2020 Quantum theory of radiation by nonstationary systems with application to high-order harmonic generation Phys. Rev. A 101 013140) where a similar approach is also developed and quantum-electrodynamical features of HHG emission are analyzed.

Reviewer #3:

Remarks to the Author:

In the last revision, the authors have taken into account my advice and included the results of the calculation of photon statistics in the HHG process. This provides an example of features treatment of which requires a formalism with a quantized description of photons, elaborated in the paper. Now the paper is more consistent and I can recommend its publication in Nature Communications.

It would be good if the authors add a brief discussion of the physical consequences of the given results on the Mandel parameter and squeezing.

Reviewer #1 (Remarks to the Author):

As before, I recommend the paper for publication. But now I strongly recommend to cite and briefly discuss one more very recent paper concerning the problem under the study (Yangaliev D N, Krainov V P, and Tolstikhin O I 2020 Quantum theory of radiation by nonstationary systems with application to high-order harmonic generation Phys. Rev. A 101 013140) where a similar approach is also developed and quantum-electrodynamical features of HHG emission are analyzed.

We thank Reviewer #1 for suggesting [Phys. Rev. A 101 013140]. We included the reference in the manuscript together with the previous [27-31] that also investigates quantum corrections of HHG process. [Phys. Rev. A 101 013140] gives a quantum-electrodynamical analysis of the spectrum of HHG however the photon statistics and other quantum features are not investigated (which are the focus of our manuscript).

Reviewer #3 (Remarks to the Author):

In the last revision, the authors have taken into account my advice and included the results of the calculation of photon statistics in the HHG process. This provides an example of features treatment of which requires a formalism with a quantized description of photons, elaborated in the paper. Now the paper is more consistent and I can recommend its publication in Nature Communications.

It would be good if the authors add a brief discussion of the physical consequences of the given results on the Mandel parameter and squeezing.W

We thank the reviewer for this comment and we added a brief discussion on the Mandel parameter and squeezing in the "Discussion":

Squeezing and non-Poissonian statistics have numerous applications in the field of sensing, high resolution imaging, weak measurements and quantum communications